Novel Systems Biology Techniques

# Lost and Found: Re-searching and Re-scoring Proteomics Data Aids Genome Annotation and Improves Proteome Coverage

Patrick Willems,[a]* Igor Fijalkowski,[a] Petra Van Damme[a]

[a]Department of Biochemistry and Microbiology, Ghent University, Ghent, Belgium

**ABSTRACT** Prokaryotic genome annotation is heavily dependent on automated gene annotation pipelines that are prone to propagate errors and underestimate genome complexity. We describe an optimized proteogenomic workflow that uses ribosome profiling (ribo-seq) and proteomic data for *Salmonella enterica* serovar Typhimurium to identify unannotated proteins or alternative protein forms. This data analysis encompasses the searching of cofragmenting peptides and postprocessing with extended peptide-to-spectrum quality features, including comparison to predicted fragment ion intensities. When this strategy is applied, an enhanced proteome depth is achieved, as well as greater confidence for unannotated peptide hits. We demonstrate the general applicability of our pipeline by reanalyzing public *Deinococcus radiodurans* data sets. Taken together, our results show that systematic reanalysis using available prokaryotic (proteome) data sets holds great promise to assist in experimentally based genome annotation.

**IMPORTANCE** Delineation of open reading frames (ORFs) causes persistent inconsistencies in prokaryote genome annotation. We demonstrate that by advanced (re)analysis of omics data, a higher proteome coverage and sensitive detection of unannotated ORFs can be achieved, which can be exploited for conditional bacterial genome (re)annotation, which is especially relevant in view of annotating the wealth of sequenced prokaryotic genomes obtained in recent years.

**KEYWORDS** *Deinococcus radiodurans*, *Salmonella*, alternative translation initiation, bacterial genome (re)annotation, chimeric spectra, riboproteogenomics, spectral re-scoring

With the exponential rise of sequenced bacterial genomes, automated genome annotation pipelines are indispensable. Despite their utility, a growing body of evidence suggests that these methods underestimate genome complexity and pose a danger of propagating biases present in current annotations (1–4). For instance, in the widely adopted NCBI prokaryotic genome annotation pipeline, protein start site annotation, sequencing errors giving rise to interrupted genes, and the delineation of open reading frames (ORFs) based on homology or *ab initio* predictions remain persistent problems (5). In addition, genomic elements such as small open reading frames (sORFs) are vastly underrepresented in current genome annotations, as emphasized by their systematic identification in recent reports studying translation in bacteria (6–8). In view of the poor agreement in annotation prediction using common bacterial genome annotation pipelines and the fact that standardization remains difficult given the biological diversity (4), resolving these annotation biases necessitates experimental studies aiming to delineate protein-coding regions. During genome reannotation efforts, confident assignment of novel protein-coding regions is crucial. Such endeavors are greatly facilitated by omics techniques such as ribosome profiling (ribo-seq). Ribo-seq provides a genome-wide snapshot of *in vivo* translation through deep sequencing of mRNA fragments covered by the actively translated ribosomes and can

Address correspondence to Petra Van Damme, petra.vandamme@ugent.be.

* Present address: Patrick Willems, VIB-UGent Center for Plant Systems Biology and Department of Plant Biotechnology and Bioinformatics, Ghent University, Ghent, Belgium.

hint at unannotated translation products (9, 10). Furthermore, proteomics can provide complementary evidence of protein synthesis by searching customized protein databases such as full genome translations (11) or *de novo*-assigned ORFs based on ribo-seq and/or sequence features (6, 12, 13). Supported by complementary ribo-seq and proteomics evidence, novel ORFs, N-terminal extensions and truncations, and wrongly annotated pseudogenes and their translation products have been reported for several bacteria (12, 13).

In terms of peptide identification, correct discrimination of true positives from false positives is complicated due to the reasonably increased database size searched in proteogenomics (14). Re-scoring of peptide-to-spectrum matches (PSMs) by using machine learning tools, such as Percolator (15), can aid in assigning correct peptide identifications. Such postprocessing analysis uses several scoring features describing the quality of the PSM. For instance, search engines such as MS-GF+ deliver an extended set of PSM features (e.g., MS-GF+ score and matched fragment peak mass deviations) that can be used by Percolator (16). In addition, the deviation of the predicted peptide retention time (RT) serves as a useful scoring feature (17). In addition to predicting the RT for a given peptide, algorithms were recently introduced that predict the intensity of peptide fragment ions with unprecedented accuracy (18–21). Comparing these predicted fragment ion intensities with those matched fragment ions aids in discriminating correct PSMs by machine learning (20–22). As such, these fragment intensity correlation features are especially useful for attaining a higher confidence for novel (i.e., database unannotated) peptide identifications (23, 24).

While fragment intensity-based correlation metrics typically increase the number of tryptic peptide identifications by 5% at a false discovery rate (FDR) significance of 0.01 (20, 21), searching spectra for evidence of co-eluting, and thus cofragmented, peptides can deliver up to 30% to 64% additional peptide identifications (25, 26). The identification potential of chimeric spectra is of course dependent on many factors, such as proteome complexity or mass spectrometry (MS) instrument settings, such as precursor isolation window and dynamic exclusion time (25). Interestingly, proteins identified solely by searching such chimeric spectra tend to display lower expression levels (25). As such, searching chimeric spectra might aid the identification of unannotated peptides typically missed in a routine data analysis workflow. Here, we describe how searching chimeric spectra with postprocessing, including MS²PIP-derived features, improves the overall proteome depth and aids in identifying hypothetical and unannotated proteins. We applied our workflow to the well-characterized human bacterial pathogen *Salmonella enterica* serovar Typhimurium and validated novel protein-coding regions with complementary ribo-seq translation evidence. We further elaborate how (ribo)proteogenomics is instrumental in reannotating ORFs, the discovery of novel ORFs across bacteria, and genome annotation in general.

## RESULTS

**Maximizing peptide identification using an iterative search strategy with Percolator postprocessing.** Label-free shotgun proteomic analyses of the *Salmonella enterica* subsp. *enterica* serovar Typhimurium strain SL1344 proteome was performed, profiling the isolated proteomes of *S.* Typhimurium cultures of three consecutive exponential growth stages in triplicate (optical densities [OD] of 0.2, 0.4, and 0.6) (see Materials and Methods). For complementary evidence of protein synthesis, we relied on our previously published ribosome profiling (ribo-seq) data acquired under similar growth conditions at an OD of 0.5 (12, 27). Since we were striving to search the full complement of possible genomic ORFs, all theoretical ORFs with a minimal length of 30 nucleotides (nt) and initiated from canonical ATG or near-cognate GTG and TTG start codons (the latter two codons are estimated to account for 14% of ORF start codons [12]) of the *S.* Typhimurium genome were *in silico* translated and used for database searching. The resulting database contains ~320,000 ORF translations, though it exhibits a high level of redundancy, as overlapping entries with in-frame translation starts are prominent and thus differ solely by their N-terminal protein sequence. To remove

this redundancy, a nonredundant tryptic peptide database of 2.2 million peptides was constructed (see Materials and Methods). Of these, approximately 550,000 peptides (25%) match annotated (Ensembl) proteins, whereas 1.6 million hypothetical peptides (75%) stem from *in silico* genome translation. Hence, searching a six-frame translation results in just a 4-fold increase in database size, in contrast to the manifold-increased sizes that result when such a rationale is applied to eukaryotic genomes. To assign contaminant peptides, 8,042 tryptic peptides of the *in silico*-digested cRAP database (25) were appended to the tryptic peptide database.

A concatenated target-decoy peptide database was searched using MS-GF+ Percolator (15, 16). Here, 23 MS-GF+-derived scoring features are used for semisupervised machine learning by Percolator to discriminate true peptide identifications. In addition to MS-GF+ features, we relied on an additional set of 11 features, referred to as the auxiliary feature set, and included the experimental RT deviation (ΔRT) from predictions (28) in addition to the number of missed cleavages, features related to the number of matched b/y ions, and features describing correlation of b/y-ion intensities to those of MS²PIP-predicted spectra (18, 22) (Fig. 1). A full description of all 34 features is provided in Table S1. To assess the merit of the feature sets, PSMs were re-scored by Percolator using either the default MS-GF+Percolator features, the auxiliary features, or the combined feature set. After postprocessing, spectral and peptide $q$ values were re-estimated separately for annotated and novel peptides based on the recalibrated Percolator scores. Importantly, this postprocessing strategy was also used for chimeric peptide identification. To this end, we implemented an iterative search strategy, similar to that of Shteynberg et al. (26), where fragment ions of a confidently identified peptide (PSM $Q$ value ≤ 0.01) are removed and the resulting "cleaned" spectrum is searched in the next search round to identify potential cofragmented peptides (see Materials and Methods). A graphical example of PSM re-scoring and re-searching of cleaned spectra is provided at https://doi.org/10.6084/m9.figshare.12847904.

First, we assessed the performance of the different feature sets by the identification of annotated (Ensembl) peptides. Using the combined feature set results in an increase of 1.49 ± 0.14% (mean ± standard deviation [SD]) nonredundant peptides identified per sample in the first search compared to default MS-GF+Percolator processing, accumulating up to 545 additional unique peptides (+2.1%) across all samples (Tables S2 and S3). Interestingly, the synergy of postprocessing by this combined feature set is more prominent for the chimeric searches. Across all samples, an additional 2,430 (+14.4%) and 2,660 (+54%) nonredundant peptides were identified using the combined feature set compared to the default MS-GF+Percolator processing in the second and third search rounds, respectively (Tables S2 and S3). Hence, MS²PIP-derived scoring features are increasingly useful for discriminating cofragmented peptides. For instance, next to MS-GF+ score, Pearson correlation to the MS²PIP-predicted spectrum is a useful independent scoring metric to distinguish high-quality PSMs (Fig. 2B). This is further supported by the learned Percolator SVM (support vector machine) weights, indicating that Pearson correlation is a useful feature for discriminating target PSMs in the first and subsequent iterative searches (spec_pearson_norm; https://doi.org/10.6084/m9 .figshare.12849851). It should be noted that high spectral correlation values can be obtained for PSMs with few matched fragment ions. This issue can be (partially) compensated for by including features related to the number of identified ions (Table S1). Taken together, the results show that re-scoring PSMs using the auxiliary feature set identified 98.5% (25,493/25,894) of the PSMs identified using default MS-GF+Percolator processing (peptide $Q$ value ≤ 1%), whereas 998 (+5.9%) and 1,174 (+23.8%) additional nonredundant peptides were identified in chimeric search rounds 2 and 3, respectively (Tables S2 and S3).

As a benchmark, we validated the performance of our pipeline by comparing identified annotated peptides with those identified by a routine MaxQuant search (Fig. 2C). The first search identified 26,440 unique peptides (peptide $Q$ value ≤ 1%), sharing 22,916 peptides (91.9%) with the MaxQuant output (Fig. 2C), and overall 1,550 nonredundant peptides more. Whereas the second and third search identified an

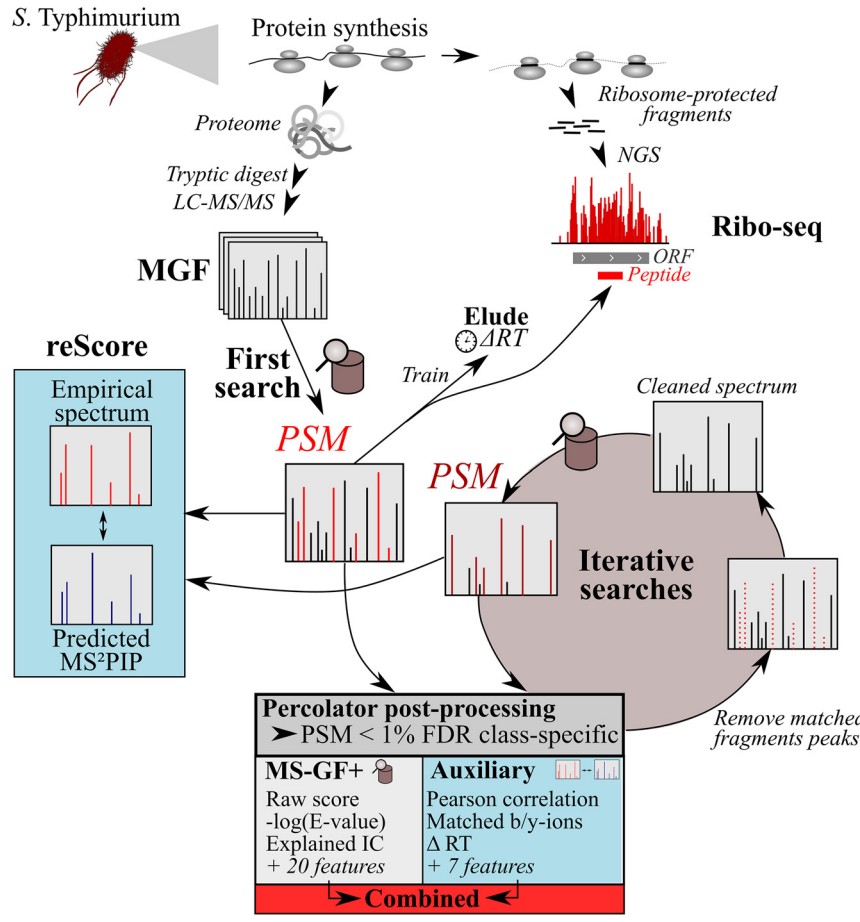

**FIG 1** Proteomics pipeline using Percolator (15) postprocessing of 34 features (Table S1). *S.* Typhimurium protein expression was studied by ribosome profiling (ribo-seq) data (12) and proteomic shotgun analysis. Spectra were searched by MS-GF+ against an *in silico*-digested tryptic peptide database (see the text). The retention times (RT) of the top-scoring 1,000 nonredundant peptides (highest MS-GF+ score) were used to train an RT model with ELUDE (28) and calculate the deviation of empirical and predicted RT (ΔRT). Besides ΔRT, an additional 10 PSM quality features were measured, constituting the auxiliary feature set. The MS-GF+, auxiliary, or combined feature set was used by Percolator (15) for re-scoring of PSMs. *Q* values were re-estimated in a class-specific manner for annotated and novel peptides. Identified fragment ions were removed from spectra with a significant PSM (*Q* value < 0.01; combined feature set) and searched iteratively to identify cofragmented peptides. Per search, identical search and postprocessing steps were repeated as for the first search, except that the trained RT model was used from the first search and a wider precursor mass tolerance was applied (as described by Shteynberg et al. [26]).

additional 1,572 nonredundant peptides compared to the first search, the majority of peptides (78.6%) identified in the iterative searches were also identified in the first search. Interestingly, 626 peptides identified exclusively in the second search were also identified by MaxQuant. It is important to note that the "second peptide search" option is enabled by default in MaxQuant, which performs a similar iterative search of a "cleaned" spectrum for MS/MS spectra that potentially contain cofragmented peptides (29). Running an identical MaxQuant search without this second peptide search option shows that 678 cofragmented peptides were identified by MaxQuant (Table S4; https://doi.org/10.6084/m9.figshare.12850046). Interestingly, whereas 157 cofragmented peptides were identified in our first search, 411 of them were similarly identified only by our chimeric searches (https://doi.org/10.6084/m9.figshare.12850046). Taken together, the results show that compared to routine database searching, our implemented iterative search strategy combined with Percolator postprocessing is able to increase the number of confident peptide identifications.

**Searching of chimeric spectra improves identification rate of low-abundance proteins.** Identification of cofragmented peptides can increase the confidence of

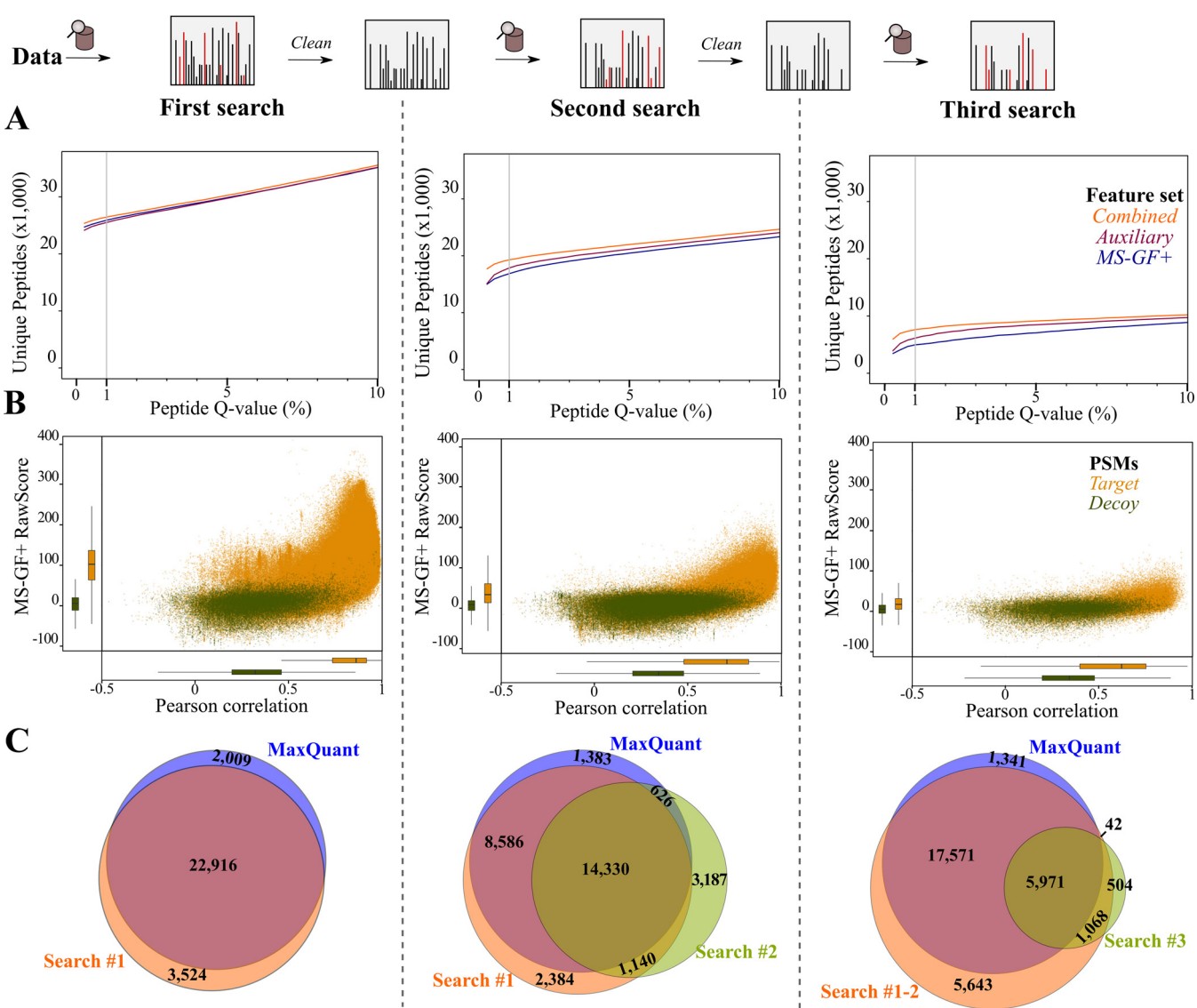

**FIG 2** Annotated peptide identification using a chimeric postprocessing pipeline. (A) Number of nonredundant peptide identifications (*y* axis) at Percolator peptide *Q*-value thresholds (*x* axis) in the first (left), second (middle), and third (right) searches. Percolator was run in parallel using the default MS-GF+ features (blue), the auxiliary features (purple), and the combined feature set (orange). (B) Scatterplot of MS-GF+ RawScore and Pearson correlation (spec_pears_norm by reScore [22]) for PSMs in the three iterative search rounds. Only features for the PSM with the highest Percolator-recalibrated score were displayed. (C) Overlap between peptides identified by MaxQuant and the three search rounds of the proteomics pipeline (combined feature set; peptide *Q* value ≤ 0.01).

protein identifications by increasing protein coverage. For instance, 3,100 proteins were identified based on the 26,440 nonredundant peptides identified in the first search round (combined feature set) (Table S3). Of the 2,175 cofragmented peptides exclusively identified in chimeric searches and identified in at least two samples (peptide *Q* value ≤ 0.01), 94.9% (2,064 peptides) matched these 3,100 protein accessions identified in the first search round. Hence, chimeric searches can identify additional peptides and improve the coverage of proteins identified in the first search round. Furthermore, the chimeric searches identify 111 additional peptides mapping to 102 proteins. Interestingly, searching chimeric spectra has previously been reported to result in the detection of previously unidentified proteins with lower mRNA-seq expression (25). We aimed to test this hypothesis by using the complementary ribo-seq data. First, we confirmed that ribo-seq expression positively correlated with protein abundance (Pearson $R = 0.43$; $P < 2.2$ e$-16$), for 2,568 nonambiguous proteins quantified by MaxQuant (Fig. 3A; Table S5). We indeed observed that the translation level of the 102 protein accessions

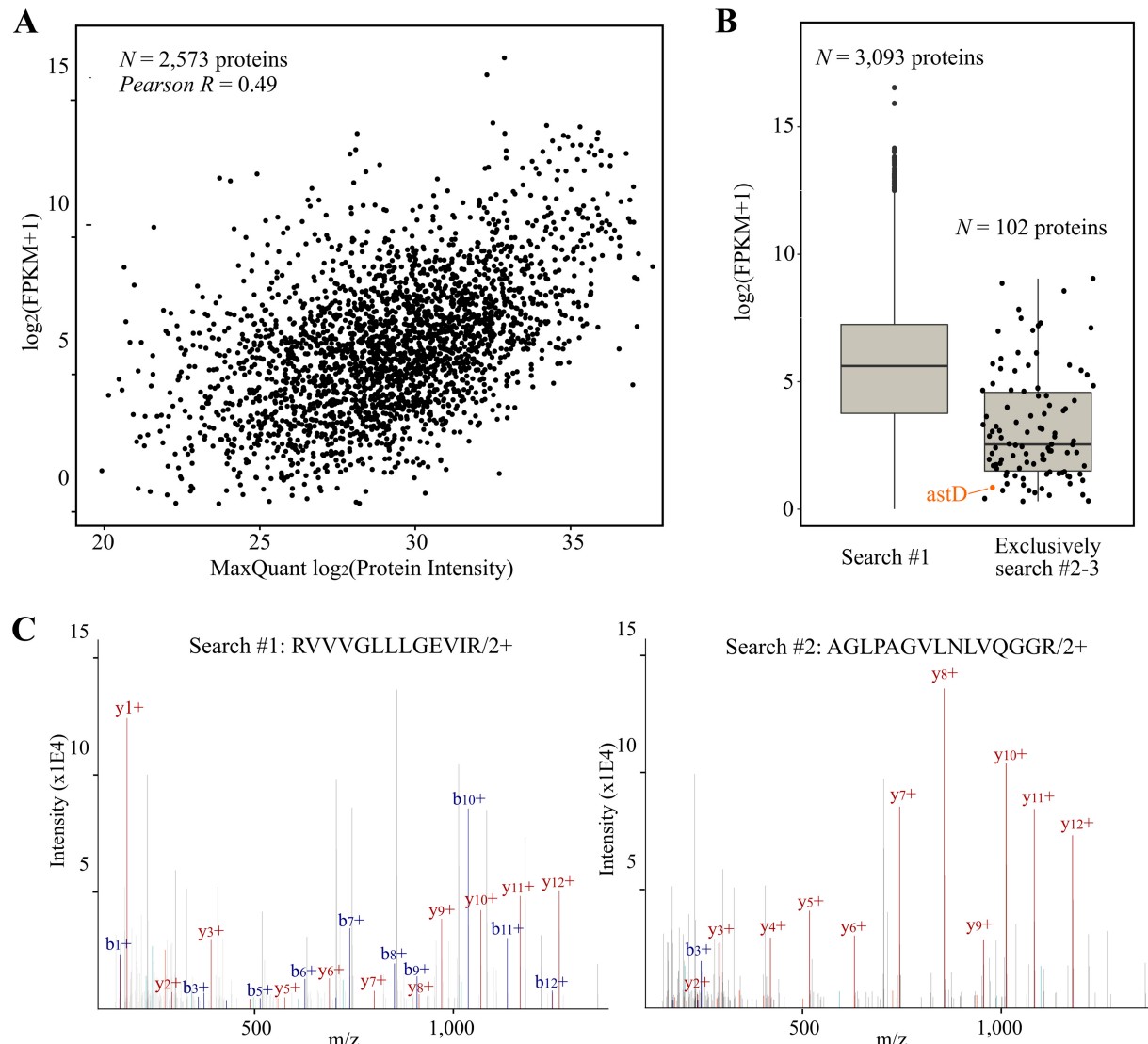

**FIG 3** Chimeric searches improve detection of low-abundance proteins. (A) Pearson correlation ($r = 0.49$) of protein abundance (MaxQuant $\log_2$ protein intensity [$x$ axis]) and ribo-seq translation levels ($\log_2$ FPKM + 1 [$y$ axis]). A total of 2,573 proteins were plotted (Table S5). (B) Ribo-seq translation levels for proteins matched by at least one unique peptide in the first search (including ambiguous peptide-to-protein assignments) (left) and for proteins exclusively identified in the chimeric searches by at least one unique peptide in at least two samples (right). The low-abundance AstD protein is indicated in orange (ribo-seq FPKM, 0.80). (C) Annotated MS/MS scan from the doubly charged RVVVGLLLGEVIR peptide identified in the replicate 1 sample at an OD of 0.8 in the first search round (left) and the double-charged AGLPAGVLNLVQGGR peptide in the second search round (right). Matched b/y ions are indicated in blue and red, respectively.

exclusively identified in chimeric searches by our pipeline (≥2 samples) shows lower translation evidence than proteins identified in the first search round (Fig. 3B; Table S6). As such, chimeric searches facilitate the detection of proteins with lower abundance. For instance, *astD*, a gene encoding a metabolic enzyme involved in Arg degradation, is poorly translated (ribo-seq FPKM [fragments per kilobase per million] = 0.80) and is solely matched by the peptide AGLPAGVLNLVQGGR in iterative search rounds. Note that this peptide and protein were also identified and quantified as low abundance by MaxQuant ($\log_2$ protein intensity = 22.84) when the "second peptide search" option was enabled (Table S4). More precisely, this peptide was identified in 8 of 9 samples at a nearly identical elution time (~106.6 min) with the cofragmenting peptide RVVVGL LLGEVIR (originating from trigger factor protein) identified in the first search round (Fig. 3C). Moreover, AGLPAGVLNLVQGGR was also identified twice in the third iterative chimeric search, after identification of RVVVGLLLGEVIR in the first round, and still

another cofragmented peptide, SAEALQWDLSFR (RNase E), was identified in the second search round (https://doi.org/10.6084/m9.figshare.12850142). Taken together, the results show that iterative searching of MS/MS spectra improves the protein coverage for proteins and facilitates the detection of proteins with low abundance.

**Proteogenomics.** In addition to searching annotated peptides, our custom peptide library comprised ~1,650,000 unannotated tryptic peptides (75% library) derived from *in silico* translation of ORFs of at least 30 bp. Confidence of proteogenomic-indicative peptides was assessed in a class-specific manner to conform to minimum guidelines proposed for proteogenomics (14). By using the postprocessing strategy described for annotated peptides, 147 and 53 peptides were identified at a 5% peptide $Q$ value by the combined feature set in the first and second searches, respectively (Fig. 4A; Table S7). Note that at a 5% peptide $Q$ value, no novel peptides were identified in the third search round. Similar to the annotated peptide searches, the combined feature set improves the identification rate compared to the default MS-GF+ features (first search, +24.6% peptides; second search, +60.6% peptides) (Fig. 4A). Whereas the MS-GF+ score and MS²PIP-derived Pearson correlations show similar distributions for target and decoy peptides, both scoring metrics aid in distinguishing high-scoring PSMs (Fig. 4B). When the distributions of MS-GF+ score, Pearson correlation, and the fraction of explained ion current for PSMs matching annotated and novel peptides ($Q$ values of 1% and 5%, respectively) were compared, similar distributions could be observed (Fig. 4C). Hence, these PSM quality metrics indicate that the quality of spectral matches to novel peptides is similar to that of annotated peptides.

As another independent quality metric, we considered the ribo-seq coverage the genomic region encoding the respective peptide (Fig. 4D), distinguishing peptides showing strong translation (ribo-seq RPKM > 10), low translation (ribo-seq RPKM < 10), or no ribosomal footprints. It can clearly be observed that the stringent class-specific 5% FDR scoring leads to the identification of novel peptides with strong translation evidence. More than 80% of the genomic regions encoding novel peptides identified in the first and second search have a ribo-seq RPKM of >10, which is similar to the proportion in annotated peptides (Fig. 4D). In addition to class-specific FDR scoring of novel peptides, we also performed Percolator postprocessing on the novel and annotated peptides together, filtering at a 1% FDR. As anticipated, this greatly increased the number of novel peptides, though only 31% to 36% of peptides corresponding to strongly translated genomic regions were retained (Fig. 4D). As such, strict class-specific FDR scoring of novel peptides is strongly recommended to deliver high-confidence identifications, as global FDR scoring together with annotated peptides likely underestimates the FDR of novel peptides.

In the next phase, we set out to inspect the identified peptides indicative of novel protein-coding regions in *S.* Typhimurium. To generate our final peptide candidate list, we considered the 193 unannotated peptides identified at a 5% peptide $Q$ value threshold using either one of the feature sets (Table S7; Fig. 5A, top). In total, 942 MS/MS spectra of representative PSMs for the 193 unannotated peptides (PSMIds in the Percolator peptide-level reports) were obtained (https://doi.org/10.6084/m9.figshare.12852641). Whereas 165 of 193 peptides (85.5%) are identified after re-scoring with the combined feature set, the MS-GF+ and auxiliary feature sets deliver an additional 28 nonredundant peptides. However, the combined feature set delivers 38 extra unannotated peptides. For instance, in the case of the peptide SSLLSTHK, matching an N-terminal extension of the annotated *yccA* with an estimated peptide $Q$ value of 3.45% using the combined feature set, the same PSM had a peptide $Q$ value of 11.5% using the default MS-GF+ processing (Fig. 5B). Here, the auxiliary features used, e.g., a Pearson correlation of 0.92 and a full series of identified y ions, aided in assigning this peptide as a significant hit. Assigning peptide identifications per ORF reveals 31 unannotated ORFs matched by at least two unique peptides, 49 ORFs matched by one unique peptide with at least 2 PSMs, and 48 ORFs matched by a peptide with a single PSM (Fig. 5A, bottom). To restrict our manual curation effort, we ignored ORFs matched

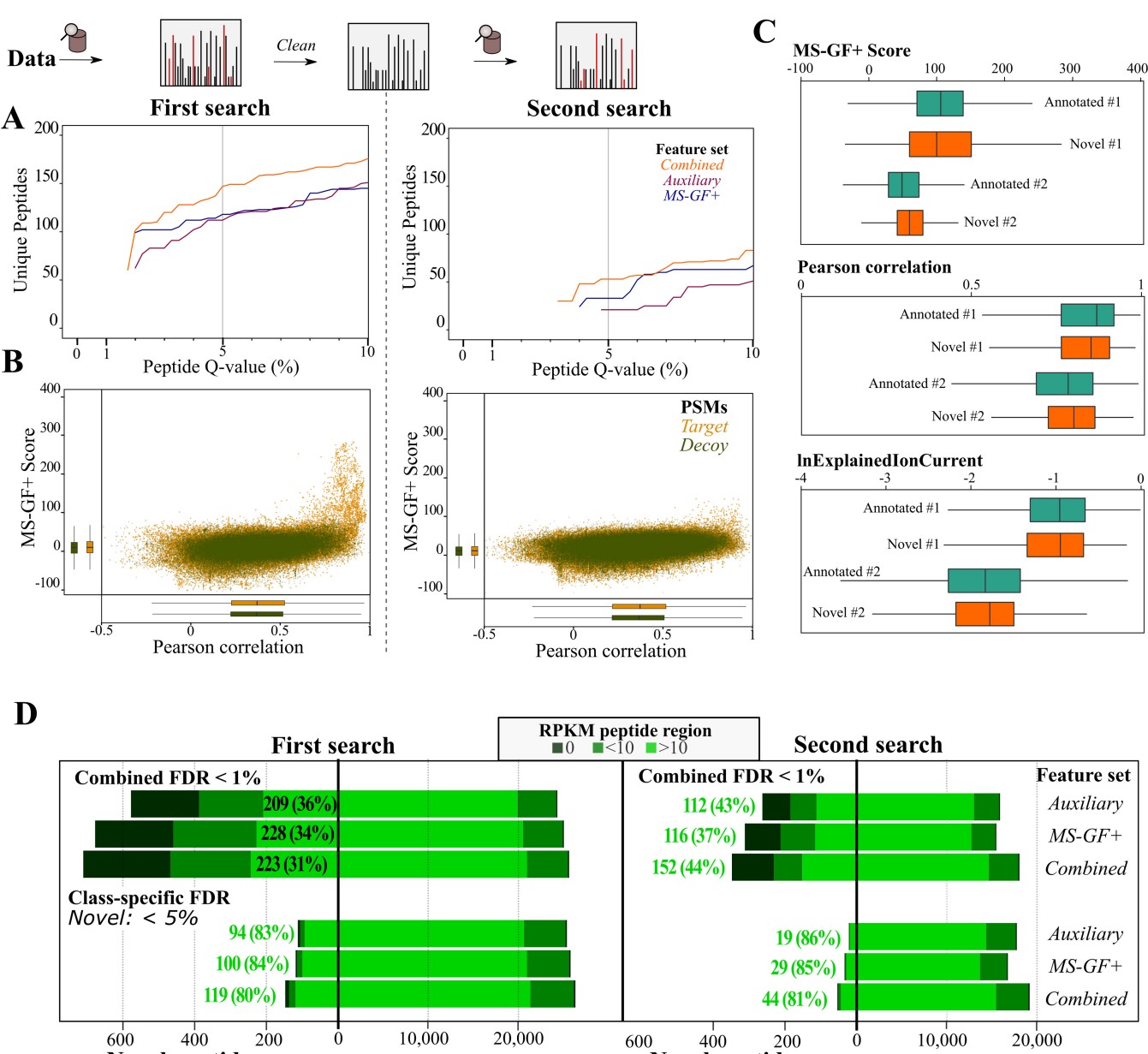

**FIG 4** Unannotated peptide identification using a chimeric postprocessing pipeline. (A) Number of nonredundant peptides (*y* axis) at Percolator peptide *Q*-value thresholds (*x* axis) in the first and second searches. Percolator was run in parallel using the default MS-GF+ features (blue), the auxiliary features (purple), and the combined feature set (orange). (B) Scatter plot of MS-GF+ RawScore and Pearson correlation (spec_pears_norm by reScore [22]) (Table S1) for PSMs in the two search rounds. Only features for the PSM with highest Percolator recalibrated score after postprocessing using the combined feature set are shown. (C) Distributions of MS-GF+ score, Pearson correlation, and logged explained ion current (lnExplainedIonCurrent) distribution for PSMs with *Q* values below 1% (combined feature set) for annotated peptides (green) or below 5% for unannotated peptides (orange). (D) Ribo-seq coverage for annotated and novel peptides identified in the first and second searches using different feature sets for combined FDR or class-specific FDR estimation. Ribo-seq reads per kilobase of transcript per million reads mapped (RPKM) were calculated for genomic regions encoding the respective peptide, distinguishing highly translated regions (RPKM > 10), low-translated regions (RPKM < 10), and peptide genomic regions without ribosome footprints (RPKM = 0).

by only a single PSM, leaving 80 ORFs supported by at least two PSMs. After inspection of PSMs in the context of corresponding ribo-seq coverage and *de novo* ORF predictions (12) in a genome viewer, we categorized 66 of 80 novel ORFs (82.5%) as high confidence. More specifically, when classifying these ORFs with respect to Ensembl-annotated ORFs, we identified 15 novel intergenic ORFs, 1 novel ORF mapping to a so-called noncoding RNA, 38 N-terminal extensions, six N-terminal truncations, three ORFs located at pseudogenes, two frameshifts, and one example where a peptide

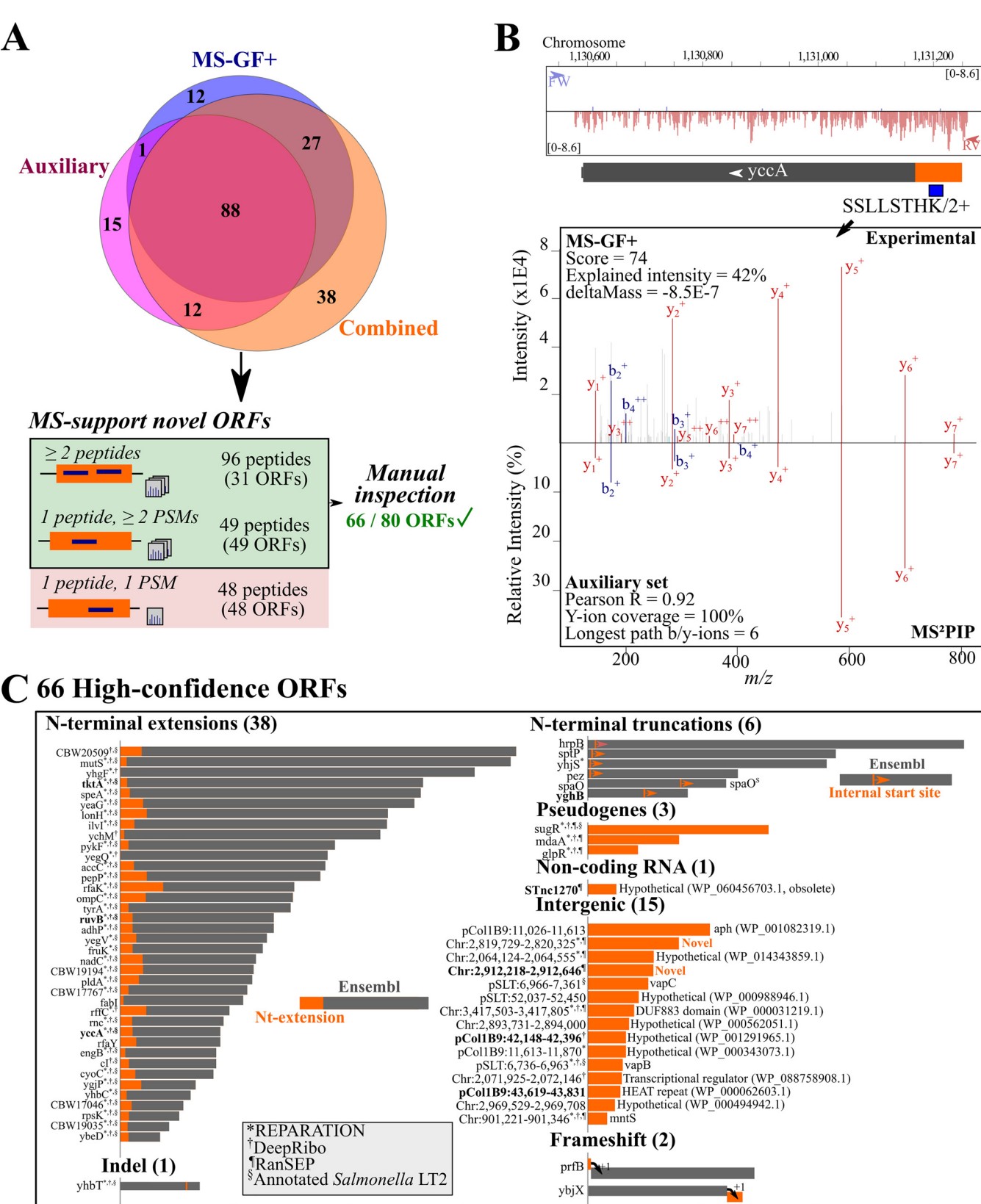

**FIG 5** *S.* Typhimurium unannotated protein-coding regions. (A) (Top) Venn diagram of unannotated peptides identified at a peptide *Q* value of ≤0.05 after Percolator processing using the MS-GF+, auxiliary, or combined feature sets. (Bottom) Peptide-to-ORF assignment, resulting in 66 high-confidence ORFs after

overspanned an annotated amino acid deletion in the interrupted *yhbT* protein-coding region (Fig. 5C). Notably, 7 of 66 high-confidence ORFs were matched solely by peptides identified using the auxiliary and/or combined feature sets (Fig. 5C, bold). This included, for instance, the N-terminal extension of *yccA* matched by the peptide SSLLSTHK (Fig. 5B). In addition, the N-terminal extension of RuvB was matched only by the cofragmented peptide LLAEYVGQPQVR (see https://doi.org/10.6084/m9.figshare.12852641 for the annotated spectrum), due to iterative searching implemented in the pipeline. Taken together, these results convincingly demonstrate that more advanced re-scoring and re-searching directly aid in novel ORF delineation.

Of the 38 N-terminal extensions with matching peptide evidence, 33 (87%) were previously predicted by REPARATION, and for 14 of these, matching peptide evidence was previously reported (12). However, we now report peptide evidence for an additional 24 N-terminal extensions with one (14) or more (10) matching peptides. Similarly, we identified six peptides hinting at N-terminally truncated protein variants or proteoforms. This includes, for instance, the functionally characterized internally translated short SpaO proteoform (SpaO^S) (30), which was originally described in the *Yersinia* homologue *yscQ* (31). In addition, there was proteomic evidence for a putative internal translation site in the case of *yghB*, suggesting a putative strong truncated proteoform. The other four N-terminal truncations are truncations of a few amino acids, of which two were predicted by REPARATION (12). Note that unlike N-terminal extensions, putative truncations are discernible only via the N-terminal peptide, where a noncanonical start codon encodes an initiator Met (and thus not Val [GTG] or Leu [TTG]).

We also controlled whether the 66 high-confidence novel ORFs were annotated in the closely related *S.* Typhimurium LT2 model strain. To this end, we performed a genome alignment using Mauve (32) (https://doi.org/10.6084/m9.figshare.12850148, panel A) and inspected corresponding ORFs. Remarkably, 32 of 38 extensions (but no truncations) were correctly annotated in strain LT2, suggesting probable misannotations in the case of SL1344 (Fig. 5C, section sign [§]). For instance, the VapB and VapC proteins on the virulence pSLT plasmid were correctly annotated in strain LT2 (https://doi.org/10.6084/m9.figshare.12850148, panel B). Notably, a 42-amino-acid intergenic ORF (Chromosome:901,221-901,346) is experimentally characterized as encoding the manganese transporter protein MntS in *E. coli* (33). However, the corresponding genomic region is lacking annotation in both *S.* Typhimurium SL1344 and LT2 (https://doi.org/10.6084/m9.figshare.12850148, panel C). A similar annotation conflict is the frameshift in the gene encoding polypeptide chain release factor 2 (*prfB*), which is experimentally characterized in *Escherichia coli* (34) but lacking annotation in *S.* Typhimurium (https://doi.org/10.6084/m9.figshare.12850376, panel A). More specifically, three peptides matched a short ORF overlapping the N terminus of PrfB in another reading frame (https://doi.org/10.6084/m9.figshare.12850376, panel A). Interestingly, the peptide FRPHSNANPPGPAPAKPR possibly reflects an unannotated frameshift event in the case of *ybjX*, reconstituting almost identical *ybjX* database protein entries of other *Salmonella enterica* strains (e.g., AZT64815.1 and RXQ36930.1) (https://doi.org/10.6084/m9.figshare.12850376, panel B). It was identified in two of three OD 0.4 replicates at an identical elution time. Due to two missed cleavages, likely due to less efficient cleavage before Pro, its peptide precursor has a 4+ charge state, and several doubly charged fragment ions are observed (https://doi.org/10.6084/m9.figshare.12850376, panel B).

In the case of the three ORFs located at pseudogenes, strong peptide evidence was provided before for SugR matching protein-coding *sugR* in *S.* Typhimurium LT2 (12).

**FIG 5** Legend (Continued)
manual inspection (see Materials and Methods). (B) (Top) Annotated MS/MS spectrum of the doubly charged SSLLSTHK; (bottom) MS²PIP-predicted MS/MS spectrum. Features used for Percolator postprocessing are displayed. (C) Overview of 66 high-confidence unannotated protein-coding regions. Bars are indicative of protein size; gray indicates Ensembl-annotated regions, whereas orange indicates unannotated protein regions. The corresponding Ensembl annotations are indicated on the left, whereas for intergenic ORFs, chromosomal locations and identical proteins identified by protein BLAST are displayed. In addition, whether ORF delineation corresponds to *de novo* predictions of REPARATION (12) (*), DeepRibo (13) (†), ranSEP-predicted ORFs (ranSEP score ≥ 0.5 [6]) (¶), or matched *S.* Typhimurium str. LT2 annotation (§) is indicated. Eight translation products identified only by peptides due to re-scoring and/or iterative searching are indicated in bold. Nt, N-terminal.

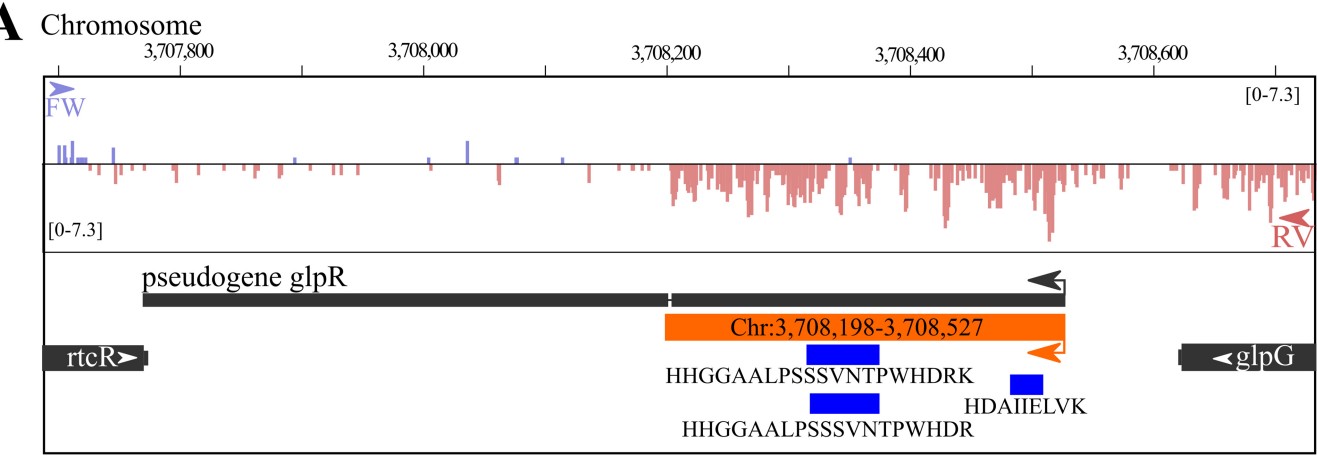

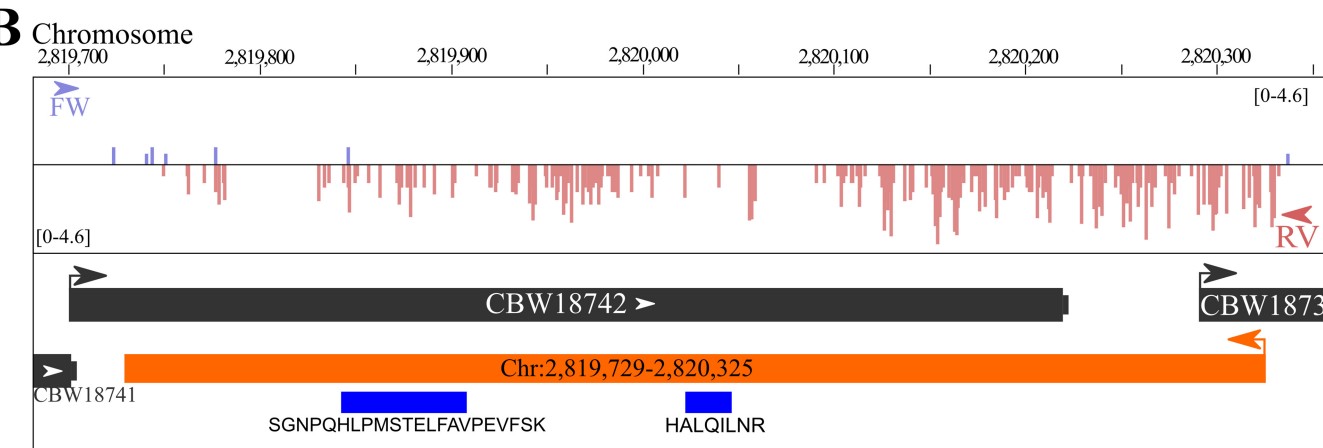

**FIG 6** Integrated genomics viewer (IGV) (58) genome view of ribo-seq read density and identified unannotated peptides for Chr:3,708,198-3,708,527 (A) and Chr:2,819,729-2,820,325 (B).

Whereas *mdaA* and *glpG* are also protein coding in *S.* Typhimurium LT2, our ribo-seq and proteomics evidence suggests truncated forms in *S.* Typhimurium SL1344. For instance, a drastically shortened GlpR protein (110 amino acids) is translated, of which the initial 107 amino acids equal the N-terminal region of the 256-amino-acid *S.* Typhimurium LT2 GlpR protein (Fig. 6A). Another interesting case was a peptide matching the "noncoding RNA" STnc270, suggesting a protein that was identical to a hypothetical protein record in *Salmonella enterica* (EHC40247.1). In addition, other protein BLAST searches against the bacterial NCBI RefSeq database revealed entries to be identical to 10 unannotated ORFs identified here besides VapB, VapC, and MntS (https://doi.org/10.6084/m9.figshare.12850148, panels B and C). All 10 are redundant, so-called "multispecies" protein entries, of which 7 hypothetical proteins were predicted by gene prediction algorithms. Hence, many of these hypothetical ORFs likely are *bona fide* protein-coding genes in *S.* Typhimurium and perhaps several other bacterial species. Interestingly, two remaining intergenic ORFs did not correspond to any protein. However, the longer one, intergenic Chr:2,819,729-2,820,325 ORF encoding a 199-amino-acid protein, shared 70% identity with a hypothetical protein in *Klebsiella pneumoniae* lacking any known functional protein domains (WP_142762344.1). Furthermore, strong complementary evidence of Chr:2,819,729-2,820,325 translation is provided for this ORF by ribo-seq and proteomics (Fig. 6B). Besides a strong ribosome density-matching translation of the newly identified ORF on the reverse strand of the annotated CBW18741 gene, two unannotated peptides were identified: HALQILNR was identified in six of nine samples, and one PSM was found for the second peptide, SGNPQHLPMSTELFAVPEVFSK (Fig. 6B).

Overall, the proteomic data sets covering 3 different (closely related) growth conditions resulted in the identification of 3,202 of 4,670 annotated *S.* Typhimurium SL1344 proteins (69% coverage), thereby reproducing the translated proteome as deduced from ribosome profiling (12) and thus making it one of the most highly covered *S.* Typhimurium proteomic data sets reported to date. However, since missing annotation can be addressed further by sampling growth conditions characterized by diverse protein expression profiles (e.g., diverse growth phases and environmental stresses), we ran the automated pipeline on an offline fractionated sample obtained after the equal mixing of *S.* Typhimurium proteomes grown under 10 (infection-relevant) conditions, as reported in reference 35. In total, 49 novel peptides were identified upon re-searching and re-scoring with the combined feature set (Table S7C). Whereas 40 of these novel peptides were found in the above-mentioned proteogenomic analysis, 9 additional novel peptides, of which 6 provided additional peptide support for high-confidence ORFs, were identified (Table S7). The extra 3 novel peptides are suggestive of alternative proteoform expression in the case of *lon*, *rof*, and *psiA* translation. In case of *rof* and *psiA*, translation of the N-terminally extended gene could again be confirmed by matching ribo-seq data (12, 27). In summary, diverse proteome sampling can aid proteome coverage and improve bacterial genome annotation.

**Proteogenomics test case: *Deinococcus radiodurans*.** Given the expanding number of sequenced bacterial genomes, a multitude of genome annotations are simply propagated. Given certain differences in annotation pipelines, e.g., different gene prediction algorithms and manual curation efforts, specific ORFs may be consistently wrongly annotated or unannotated. Proteogenomics can be a powerful tool to correctly delineate ORFs, next to the identification of novel ORFs, as shown here in the case of *S.* Typhimurium. Interestingly, for diverse bacterial species, a vast and expanding resource of proteomics data is accessible through public repositories for proteomics data, such as PRIDE (accessed June 2019) (36) (Fig. 7). Next to *Salmonella* (26), *E. coli* (>100 data sets) and *Pseudomonas* species (81 studies) represent well-studied bacterial species in PRIDE. However, several other less-studied clades sometimes have high-quality shotgun data sets available and might be readily used for proteogenomic purposes. For instance, we tested the general applicability of our proteogenomic pipeline to a proteomic data set of *Deinococcus radiodurans* strain R1 (PRIDE accession no. PXD011868) (37) (Fig. 7, bold). This bacterium belongs to the phylum *Deinococcus-Thermus*, whose members are highly resistant to extreme environmental conditions but are relatively underrepresented in terms of proteomic data sets available (at least four data sets, accessed June 2019). Despite this, extremophile bacteria such as *Deinococcus radiodurans* are gaining interest for biotechnological applications due to their high resistance to UV radiation, desiccation, and oxidative conditions (38), among others. In addition, in contrast to the current *E. coli* paradigm for bacterial translation that an mRNA includes an untranslated leader sequence (UTR) harboring the ribosome-binding site (RBS) upstream of the translation initiation codon, *Deinococcus* species show a remarkably high proportion of leaderless mRNAs lacking 5′ UTRs. More specifically, up to 1,174 leaderless transcripts (encoding 60% of genes) were expressed in *Deinococcus deserti* (39), whereas bioinformatics analysis suggested more than 40% leaderless mRNAs in *Deinococcus radiodurans* (40). Thus, proteogenomics efforts in *Deinococcus* may inform on alternative modes of bacterial translation.

Here, we reanalyzed a proteomic data set studying the response to simulated vacuum conditions (37). As additional metadata, we used a recent mRNA-seq data set studying the effect of hydrogen peroxide (41). *Deinococcus radiodurans* strain R1 contains two chromosomes, a megaplasmid, and a small plasmid, totaling 3,284,156 bp (42) (a 35% reduction compared to the *S.* Typhimurium SL1344 genome size). Applying our described proteogenomic pipeline on a 6-frame-translation database as described for *S.* Typhimurium (Fig. 1; also, see Materials and Methods), we identified 1,071 Ensembl-unannotated peptides at a 5% peptide *Q* value using the combined feature set, of which 101 were identified only as cofragmenting peptides in search round 2 or

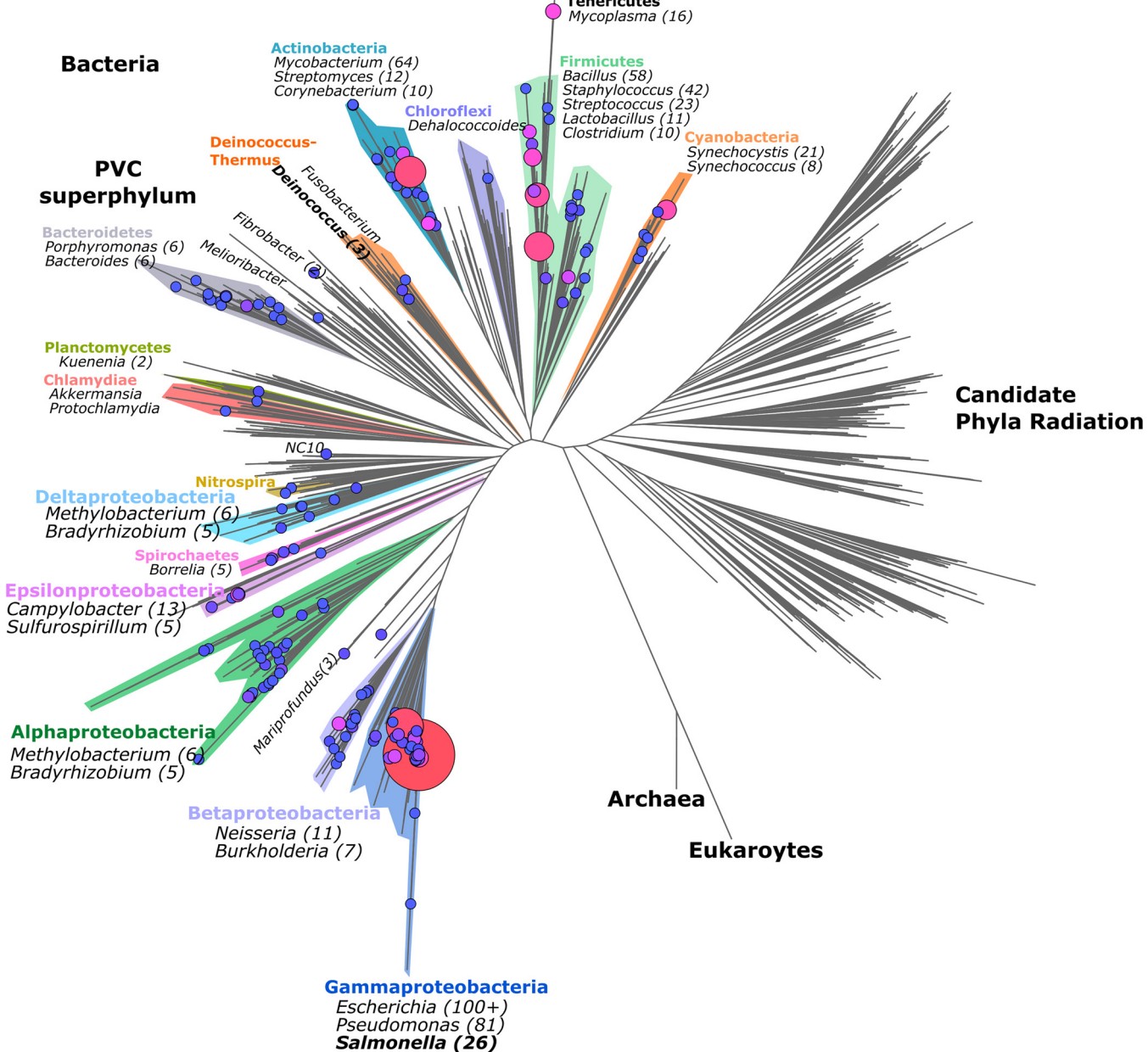

**FIG 7** Proteomic shotgun data sets across the bacterial phylogeny available in the PRIDE repository (36). The bacterial phylogeny was adapted from the work of Hug et al. (59), omitting the *Archaea* and *Eukarya* for ease of visualization. PRIDE accession records of bacteria were retrieved and catalogued using NCBI taxonomy identifiers. Proteomic data set identifiers plotted are given in Table S8 with node sizes corresponding to the number of available data sets and plotted using FigTree (version 1.4.3. [2009]; http://tree.bio.ed.ac.uk/software/figtree/).

3 (Table S9). Notably, 273 of 1,071 (25.5%) were annotated in the NCBI RefSeq proteome annotation (ASM856v1) of this strain but lacking in the Ensembl annotation. Assigning these 798 NCBI and Ensembl-unannotated peptides to the longest *in silico*-translated ORF yields 59 putative novel ORFs supported by at least two peptides (Table S9). These 59 putative ORFs match nine N-terminal extensions (15%) (Fig. 8A), a smaller proportion compared to our proteogenomic results obtained for *S.* Typhimurium (38/66 [58%] unannotated ORFs).

Thanks to the high occurrence of leaderless mRNAs, the mRNA-seq coverage can provide direct complementary evidence of probable translation start sites in certain cases. For instance, in the case of the annotated ORF DR_1392, translation (and transcription) starting from an upstream canonical start codon is supported by two

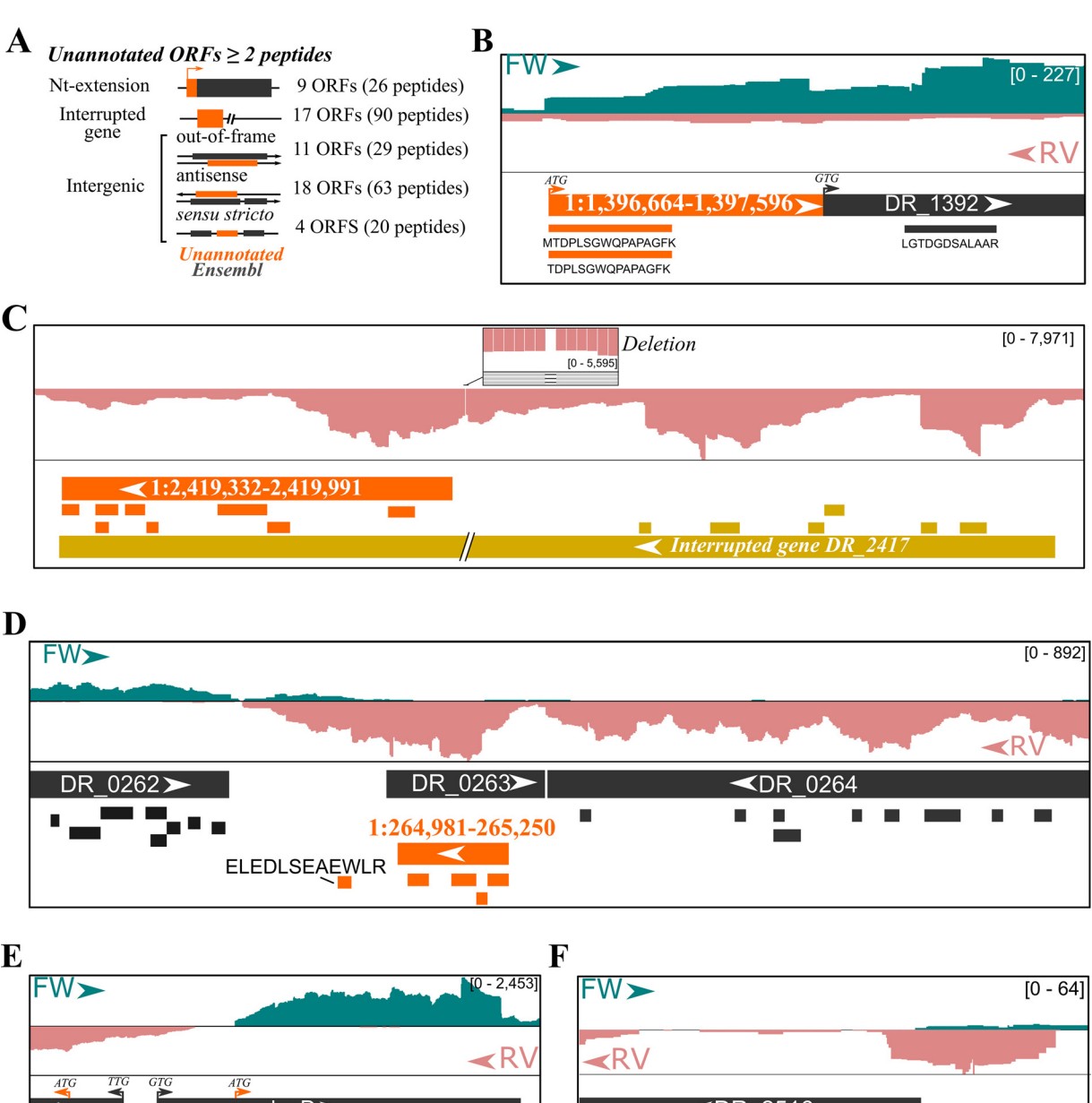

**FIG 8** Putative ORFs in *Deinococcus radiodurans* strain R1 with matching proteogenomic evidence. (A) Categorization of 59 putative ORFs with at least two matching peptides not present in Ensembl or NCBI annotation (assembly ASM856v1). (B to F) Genome view by IGV (58) showing identified annotated (dark gray) or novel (orange) peptides and their matching (longest) ORFs. Putative novel translation start sites are indicated by arrowheads labeled with the respective start codon. In addition, stranded mRNA-seq coverage was displayed from unstressed wild-type *D. radiodurans* strain R1 (41). Genome region coordinates are 1:1,396,647-1,396,869 (B), 1:2,419,754-2,421,036 (C), 1:264,080-266,689 (D), 1:2,162,416-2,163,711 (E), and 1:2,518,399-2,519,021 (F).

N-terminal peptides (with and without initiator methionine [iMet] processing) and clear evidence of leaderless mRNA-seq expression (Fig. 8B). In addition, 17 putative ORFs were located in genomic regions annotated as containing frameshifts (Fig. 8A). However, the unannotated ORFs identified here clearly result from the incorrect annotation of a gene interruption due to sequencing errors. For instance, the ORF 1:2,419,332-2,419,991 had eight matching peptides, whereas upstream novel peptides were also identified in an authentic frameshift region (Fig. 8C). mRNA-seq coverage clearly

indicates a base pair deletion (Fig. 8C, zoomed region), giving rise to an interrupted coding sequence (CDS). In fact, this sequencing error was pointed out earlier experimentally, and the correctly reconstituted DncA was characterized as an essential nuclease involved in growth and radiation resistance of *Deinococcus radiodurans* (43).

Finally, the remaining 33 putative ORFs were intergenic with reference to Ensembl-annotated ORFs (Fig. 8A), i.e., either overlapping annotated out of frame (11 ORFs) or antisense (18 ORFs), or nonoverlapping between annotated ORFs (4 ORFs). For instance, the 90-amino-acid-encoding ORF 1:264,981-265,250 was matched by four peptides and matching mRNA-seq expression evidence (Fig. 8D). In contrast, the predicted antisense DR_0263 ORF lacked any peptide match or mRNA-seq evidence. In this case, DR_0263 could thus potentially be a false prediction, with DR_0262 as the last protein-coding ORF of the sense transcript and the novel 1:264,981-265,250 ORF as part of an antisense transcript following DR_0264. Moreover, the novel peptide ELEDLSEAEWLR (Fig. 8D) suggests another antisense 82-amino-acid-protein-coding ORF (1:264,758-265,003; one peptide match) on this transcript, or alternatively a frameshifted protein translated from these two neighboring ORFs. For 16 of 18 "antisense" ORFs, no supporting mRNA-seq or proteomic evidence is apparent for the annotated ORF, suggesting incorrect gene predictions.

In the two other cases where annotated ORFs have matching evidence, peptide and mRNA-seq coverages suggest nonoverlapping truncated forms of both ORFs to be a more likely scenario. For instance, several peptides matched the annotated *deoD*, where four novel peptides matched the putative antisense ORF 2:264,981-265,250 (Fig. 8E). Closer inspection of mRNA-seq coverage suggests that matching leaderless mRNA transcripts with canonical ATG start codons for the annotated *deoD* and the novel ORF are more likely than the ORFs corresponding with translation being initiated at the upstream, in-frame near-cognate GTG start sites, thus finally resulting in nonoverlapping protein-coding ORFs on opposite strands (Fig. 8E). In the case of out-of-frame sense overlapping ORFs, peptide evidence was discovered for both ORFs, suggesting possible partial overlaps. For instance, the putative ORF 1:2,518,632-2,518,976 likely encodes a 115-amino-acid protein from a leaderless transcript that, at the C terminus, overlaps with DR_2516 supported by a single peptide matching the annotation (Fig. 8E). However, it is not always possible to discriminate protein start sites, and further (matching) experimental evidence and manual curation would be required for the comprehensive delineation of ORFs in this species.

## DISCUSSION

Automated prokaryotic genome annotation is indispensable given the exponential increase in the number of sequenced bacterial genomes. Despite their utility, they can propagate inconsistent gene annotations across bacterial species (1, 44–46). Here, we applied an optimized proteogenomics workflow for bacteria to identify unannotated protein-coding ORFs and correct existing annotations. Proteogenomic searches require only a genome sequence, for which all possible canonical or near-cognate start codon-initiated ORFs are searched (≥30 bp). Similar annotation-less searching for such small protein products has been performed before in *Mycobacterium pneumoniae* (6). However, we performed class-specific FDR estimation of annotated and novel peptides, which is a basal guideline and essential for achieving high sensitivity in proteogenomics searches (14, 47). Based on ribo-seq coverage of corresponding genomic regions of the identified novel peptides (Fig. 4D), it is evident that class-specific FDR scoring greatly benefits the sensitive detection of highly translated peptides (>80%). Whereas a threshold of two unique peptides was proposed to filter true novel ORFs (6), high-scoring peptides nonetheless revealing true-positive ORFs were not considered further. For instance, of the 49 putative ORFs in *S.* Typhimurium matched by a single peptide (with at least 2 PSMs), 35 were labeled high confidence when additional meta-data, such as matching ribo-seq data (12), *de novo* ORF predictions (6, 12, 13), protein homology of the unannotated ORF, and quality of the corresponding fragmentation spectrum, were considered. Comparing the fragmentation of synthetic peptides

can additionally be used to validate "one-hit wonders" (48). Due to the recent development and accuracy of peak intensity prediction tools, such as MS²PIP (18) and Prosit (20)—the latter trained on synthetic peptides—the comparison of predicted and empirical peptide fragmentation spectra is automated and can deliver additional PSM scoring metrics to be used by semiautomated machine learning tools such as Percolator (20, 22). Notably, we previously used correlation to MS²PIP-predicted spectra to discriminate high-confidence unannotated peptides in *Arabidopsis thaliana* (24). In line with our previously reported findings obtained with the PROTEOFORMER pipeline (23), our proteogenomic pipeline illustrates how Percolator re-scoring increases the number and confidence of (un)annotated peptide identifications (Fig. 2 and 4).

Cofragmenting peptides are widespread in MS/MS spectra, and several algorithms have been developed to mine this traditionally untapped source of peptides. In our proteogenomic effort, we used an iterative search strategy similar to that of Shteynberg et al. (26) that removes fragment peaks identified in a previous search round (Fig. 1). Such iterative searching identified 19,283 and 7,585 nonredundant peptides in a second and third search round, respectively, when *S.* Typhimurium proteomics data were searched (Fig. 2). Comparison to the 678 peptides identified by the second peptide search by Andromeda (29) shows that 568 of 678 cofragmenting peptides (83.8%) were identified by our search strategy (https://doi.org/10.6084/m9.figshare.12850046). It should be noted that the second search in Andromeda is stricter, as it is performed only when overlapping precursor peptides are detected in three-dimensional (3D) liquid chromatography-mass spectrometry (LC-MS) maps (29), whereas in our iterative search strategy, all identified MS/MS spectra (PSM *Q* value, 1%) are re-searched. Supporting cofragmenting peptide identifications, 78.6% of the chimeric identifications matched peptides identified in the first search round. Furthermore, 2,064 of 2,175 peptides (94.9%) exclusively identified in iterative searches match proteins with supportive MS evidence obtained in the first search round, which overall indicates the power of our approach to increase proteome coverage. Furthermore, the other 111 cofragmenting peptides led to the discovery of 102 additional proteins that showed lower translation levels by ribo-seq than proteins identified in the first search round (Fig. 3B). Similar results have been reported, though based on mRNA-seq expression levels (25). As such, identification of cofragmenting peptides increases protein coverage as well as proteome depth and thus enables the additional identification of proteins with low abundance.

The occurrence of cofragmented peptides is dependent on several factors, such as sample complexity, the liquid chromatography set-up, and MS instrument settings. For instance, broadening the precursor *m/z* isolation width and/or shortening the dynamic exclusion time will favor the identification of cofragmented peptides (25). In the case of our proteomics data sets, an isolation width of 1.5 *m/z* and a 12-s exclusion time were used on a Q Exactive HF instrument. Although this was not done here, MS instrument settings can be tuned to facilitate the identification of chimeric spectra (25). Another alternative to avoid the preferential selection of more abundant peptide precursors is to operate in data-independent acquisition (DIA) mode. Here, no precursor selection is performed but consecutive *m/z* ranges are scanned and recorded in total, reducing the discrimination of low-abundance peptides and increasing reproducibility. Also in this endeavor, MS/MS spectrum prediction tools can prove resourceful for proteogenomics by creating spectral libraries that include potential unannotated peptides.

We first applied our proteogenomic pipeline to *S.* Typhimurium, for which we earlier reported ribo-seq-assisted de *novo* ORF predictions (12, 13). Our pipeline showed a drastic improvement in the proteomic detection of new protein start sites and intergenic ORFs, identifying 66 novel high-confidence ORFs. Notably, 8 of these high-confidence ORFs were identified only due to re-searching and re-scoring in the pipeline, and for an additional 13 novel ORFs, the process led to identification of additional peptides (Fig. 5C; Table S7). In our *S.* Typhimurium proteome analysis, correct protein start annotation was proven to be a major issue, as we obtained evidence that 38 N-terminally extended protein forms or proteoforms were corrected (Fig. 5). Such

correct delineation of protein start sites is a known challenge in prokaryotic annotation (5). However, comparing to the annotation of the related *S.* Typhimurium LT2 strain, 32 of 38 N-terminal extensions corresponded to annotated LT2 ORFs. It is likely that the misannotation of protein N termini in the case of SL1344 originates from historical reasons, as the genome assembly of *S.* Typhimurium LT2 used an updated gene annotation strategy including the reassessment of start predictions (49). Notably, the updated annotation of SL1344 by PGAP (version 4.4, updated October 2018) resolved these 32 protein start sites, emphasizing the need to compare genome assemblies and select the optimal one.

In addition to protein start sites, we also identified several novel ORFs. These ORFs encode hypothetical proteins predicted by PGAP but also proteins experimentally characterized before in *S.* Typhimurium or related species, for instance, the small protein MntS (https://doi.org/10.6084/m9.figshare.12850148, panel C) described in *E. coli* (33) or the VapB and VapC proteins discovered to be encoded on the virulence plasmid of *S.* Typhimurium Dublin (50) and later experimentally verified in *S.* Typhimurium (51). Notably, the small size (42 amino acids) of *mntS* is likely a determining factor hindering its automated gene annotation, as small ORFs are underrepresented in prokaryotic genomes (6). However, a 199-amino-acid ORF (Chr:2,819,729-2,820,325) could also be identified, likely missed due to its location antisense to annotated ORFs for which no translation evidence was found in our sampled conditions (Fig. 6B).

Other unannotated ORFs well supported by experimental evidence include those encoding alternative protein forms or proteoforms. For instance, the identified frameshift of *prfB* (https://doi.org/10.6084/m9.figshare.12850376, panel A) is a pioneering example of an efficient frameshift in *E. coli* (34), and an evolutionary study showed that for ~70% of 86 analyzed bacterial species, such frameshifting is conserved (52). In addition to frameshifting, alternative translation initiation sites downstream of the canonical start codon can give rise to N-terminally truncated proteoforms. For instance, identification of the well-characterized truncated form of SpaO (30, 31) was supported by an N-terminal peptide matching translation at the internal translation site, despite not being annotated in the LT2 and SL1344 genome annotations thus far. During inspection of putative ORFs, ribo-seq serves as a useful complementary evidence track, as illustrated for *mntS* (https://doi.org/10.6084/m9.figshare.12850148, panel C), the truncated *glpR* gene (Fig. 6A), the intergenic antisense ORF Chr:2,819,729-2,820,325 (Fig. 6B), and a multitude of other examples. Recent developments in studying translation initiation in bacteria will further facilitate precise N-terminal protein start site delineation, and concomitantly the identification of (alternative) N-terminal proteoforms (53). Taken together, with the recently discovered omnipresence of bacterial sORF translation, our results show that the translation of multiple proteoforms per gene and alternative translational decoding events such as frameshifting are largely neglected by current annotation pipelines and concomitantly that annotation databases are not properly equipped to handle these cases.

Given the ever-increasing wealth of accessible mRNA-seq, ribo-seq, and proteomics data, experimental data can be adopted to improve the confidence of protein and gene annotation. Proteogenomic analysis of samples collected under diverse growth conditions can deliver complementary and novel discovery of ORFs. For instance, next to our main proteogenomic analysis of *Salmonella* sampled at 3 consecutive phases of growth (OD 0.4, 0.6, and 0.8) under control conditions, running the automated pipeline on a prefractionated mixture of *Salmonella* proteomes grown under 10 different growth conditions (35) resulted in additional peptides for 6 high-confidence ORFs and 3 additional novel ORFs.

In an attempt to create a representative snapshot of the bacterial proteomic data sets captured in PRIDE (36), we plotted 924 data sets (accessed June 2018; https://doi.org/10.6084/m9.figshare.12852641) across the bacterial kingdom phylogeny (Fig. 7). We demonstrated the potential of proteogenomics in improving annotation of bacterial species based on publicly available data sets for *Deinococcus radiodurans* (37, 41). This extremophile is part of the phylum *Thermo-Deinococcus*, which is, despite its

interesting application potentials, relatively underrepresented in terms of available proteomic data sets (Fig. 7). We identified 59 high-confidence ORFs with at least two matching peptides. However, it is very likely that many ORFs with a single peptide hit are true positives, such as ELEDLSEAEWLR, which suggests an 82-amino-acid part of an antisense transcript (Fig. 8D). Interestingly, the strong tendency toward leaderless transcripts proved useful, as we observed strong mRNA-seq coverage starting at N termini of putative ORFs we identified (Fig. 8B, E, and F). In fact, this leaderless expression has previously been utilized to pinpoint unannotated start sites and to identify (small) proteins in *Deinococcus deserti* (39). We also observed several erroneous gene interruptions due to sequencing errors, such as in the case of *dncA* (Fig. 8C), which was in fact experimentally corrected (43).

Taken together, our results indicate that systematic (re)analysis of proteomic and high-throughput sequencing data sets can provide strong annotation potential. Importantly, the proteogenomic pipeline and applied class-specific FDR scoring result in high-confidence detection of unannotated ORFs, as evidenced by peptides characterized by high corresponding ribo-seq signals. This is of importance in facilitating high-confidence proteogenomic detection, which can optionally be followed up by manual curation, as performed here to emphasize the merit of our proteogenomic results. In the context of systematic studies, however, such manual curation may not be feasible, and additional arbitrary filters, such as the requirement of 2 peptides per unannotated ORF (as performed here for *Deinococcus radiodurans*), could be applied. Furthermore, inclusion of additional metadata filters based on homology to annotated ORFs or other sequence features could be used.

Applying the advanced proteogenomic pipelines as described here will facilitate the confident detection of novel peptides and could facilitate genome annotation in conjunction with automated gene prediction pipelines. Current automatic genome annotation pipelines remain strikingly oblivious to the wealth of functional omics data available in centralized and scattered databases. Introduction and wide implementation of advanced proteogenomic pipelines, such as the one described in this work, constitute a first, easily taken step aimed at tapping into this potential. Focusing such efforts on less-characterized species or phyla will be extremely useful to strengthen the annotation of reference model species that could extrapolated to close relatives. Such a holistic approach holds great promise for improving our understanding of bacterial genomes by benefiting from the complementary nature of interdependent omics data sets.

## MATERIALS AND METHODS

**Bacterial cultures.** The *S. enterica* serovar Typhimurium wild-type strain SL1344 (genotype, *hisG46*; phenotype, His⁻; biotype 26i) was obtained from the *Salmonella* Genetic Stock Center (SGSC, Calgary, Canada; catalog no. 438) (54). Single colonies were picked from LB (lysogeny broth) agar plates, inoculated in 3 ml LB medium without antibiotics in 14-ml round-bottom tubes (Falcon), and grown overnight at 37°C with agitation (180 rpm). Subsequently, replicate overnight precultures were diluted 1:100 in LB medium and grown at 37°C until the desired OD was reached. Aliquots (25 ml) of each replicate culture were pelleted (15 min, 5,000 rpm, 4°C), washed by resuspension in phosphate-buffered saline (PBS), and pelleted again. PBS was removed, and bacterial pellets were frozen in liquid $N_2$ and stored at −80°C until further processing.

**Proteomic sample preparation.** Bacterial pellets prepared as described above were resuspended in 300 $\mu$l of freshly prepared extraction buffer (50 mM ammonium bicarbonate [$NH_4HCO_3$] [pH 7.9], 4 M guanidinium hydrochloride). Subsequently, samples were mechanically disrupted through three repetitive flash-freezing cycles in liquid $N_2$ followed by sonication on ice (Branson probe sonifier; output level 4, 40% duty cycle, three times with 30-s bursts; 1-s pulses). Samples were cleared by 10 min centrifugation at 13,000 rpm at 4°C. The supernatants were collected, and the samples were precipitated overnight at −20°C with 4 volumes of −20°C acetone. The precipitated protein was collected by 15 min centrifugation at 3,500 × $g$ (4°C), and pellets were washed twice with −20°C 80% acetone and air dried upside down until no acetone odor remained. Samples were resuspended in 200 $\mu$l TFE (2,2,2-trifluoroethanol) digestion buffer (10% TFE, 100 mM ammonium bicarbonate) with sonication on ice (Branson probe sonifier; output level 10 to 15, 0.5-s pulses) until a homogenous suspension was formed. All samples were digested overnight at 37°C using MS-grade trypsin (Promega, Madison, WI; enzyme/substrate ratio of 1:100 [wt/wt]) with mixing (550 rpm). Subsequently, samples were acidified with trifluoracetic acid (TFA) to a final concentration of 0,5% and cleared by centrifugation at 16,100 × $g$ for 10 min at 4°C. Methionine residues were oxidized by the addition of $H_2O_2$ to a final concentration of 0.5%

for 30 s at 30°C. Solid-phase extraction of peptides was performed using $C_{18}$ reverse-phase sorbent containing 100-$\mu$l pipette tips (Bond Elut OMIX 100-$\mu$l $C_{18}$ tips; Agilent) according to the manufacturer's instructions. Bound peptides were eluted in LC-MS/MS vials with the maximum pipette tip volume of 0.1% TFA in water-acetonitrile, 30:70 (vol/vol). The samples were vacuum dried in a SpeedVac concentrator and redissolved in 20 $\mu$l of 2 mM tris(2-carboxyethyl)phosphine in 2% acetonitrile.

**Peptide database construction.** Six-frame translation of the *S.* Typhimurium SL1344 genome (Ensembl ASM21085v2) was performed, storing all ORFs of $\geq$30 bp initiated from ATG, GTG, and TTG (encoding the presumed initiator Met [iMet]). The resulting 324,010 ORFs were *in silico* digested by the "generate-peptides" function using the Crux toolkit (v3.2) (55), with trypsin P specificity with two missed cleavages and N-terminal methionine excision enabled. In addition, peptide length was set from 7 to 50 amino acids and peptide mass between 500 and 5,000 *m/z*. Decoy peptides were generated by random shuffling of peptide sequences while maintaining the C-terminal amino acid. In total, this delivered 2,170,335 target and 2,169,158 nonoverlapping decoy peptide sequences. Target peptides and their respective decoy peptides, matching proteins of the *S.* Typhimurium Ensembl proteome (ASM21085v2, 4,672 proteins), were labeled as annotated peptides. For comparison of database size, the same digestion rules were applied to the latest human Ensembl proteome (GRCh38 annotation).

**Postprocessing proteogenomics pipeline.** The pipeline we used performs re-scoring and re-searching of spectra by combining advanced MS/MS search strategies. Below we provide a brief description of the experimental procedures we followed; a more detailed version is available in Text S1.

**(i) First-round search and postprocessing.** Thermo RAW files were converted to MGF (Mascot generic format) peak lists using ThermoRawFileParser (56). The resulting MGF files were searched against the constructed peptide database using MS-GF+ (v2019.04.18) with enzymatic cleavage disabled and methionine oxidation as a fixed modification. Resulting Percolator (15) tab-delimited input files were used for default MS-GF+Percolator processing (for features, see Table S1). For each sample, the retention time (RT) of the top 1,000 ranked unique peptide sequences (highest MS-GF+ score) was used to train a peptide RT model by ELUDE (v3.02.1) and to calculate the deviation of the experimental RT as an additional scoring feature (28). In addition, fragment peak intensities of empirical spectra were compared with those predicted for MS²PIP-predicted spectra using the reScore algorithm (22). Of the resulting features, spec_pearson_norm, dot_prod_norm, and spec_mse were added to the auxiliary feature set. The auxiliary feature set was further expanded with six features, including the number of Arg/Lys residues (i.e., reflecting trypsin missed cleavages) in the peptide and features reflecting the number of matched b/y ions (Table S1). Using the MS-GF+, auxiliary, and combined feature sets, Percolator (v3.02.1) (15) was used to re-score PSMs for every MS-GF+ search (weights see https://doi.org/10.6084/m9.figshare.12849851). Subsequently, the Percolator-recalibrated PSM scores were used for class-specific confidence estimation for annotated (Ensembl) and unannotated peptides. Resulting peptide identifications were filtered at a 1% peptide $Q$ value for annotated peptides and 5% peptide $Q$ value for unannotated peptides.

**(ii) Iterative searches for identification of cofragmented peptides.** To identify cofragmented peptides in an MS/MS spectrum, we used an iterative search strategy resembling the rationale followed in the reSpect algorithm (26). For MS/MS spectra with an assigned PSM ($Q$ value $\leq$ 0.01) in the prior search, matching (tolerance $\leq$ 0.02 *m/z*) b/y ions were omitted from the spectrum (including double-charged fragment ions and/or ions with neutral losses). Spectra were re-searched for charge states 2+ and 3+ using MS-GF+ with parameters as described above, except that a wider precursor mass tolerance of 3.1 Da was set and isotope error was removed. This wider tolerance allows identification of non-monoisotopic peptides in the isolation window (26). PSM feature generation and Percolator postprocessing were performed as described above, except that the peptide RT model trained in the first search by ELUDE was used.

**Public data reprocessing.** Thermo RAW files from PRIDE accession no. PXD011868, a *Deinococcus radiodurans* vacuum stress response data set (37), were downloaded and processed as described above for *S.* Typhimurium. In addition, we downloaded raw sequencing data from Sequence Read Archive (SRA) study SRP057959, studying the transcriptomic response of *D. radiodurans* to hydrogen peroxide (36). Reads were aligned to the Ensembl genome and reference assembly (ASM856v1) that was used for the proteogenomics analysis using STAR version 2.7.3a (57), allowing 2 mismatches. Read pairs matching the forward strand were filtered based on flags 99 and 147 and flags 83 and 163 for the reverse strand (-f option; SAMtools).

**Supplemental figures.** Supplemental figures are available on Figshare.com.

**Data availability.** The mass spectrometry proteomics data have been deposited to the ProteomeXchange Consortium via the PRIDE (36) partner repository with the data set identifier PXD016377 (http://proteomecentral.proteomexchange.org/cgi/GetDataset?ID=PXD016377).

## SUPPLEMENTAL MATERIAL

Supplemental material is available online only.

**TEXT S1**, DOCX file, 0.02 MB.

**TABLE S1**, PDF file, 0.3 MB.

**TABLE S2**, PDF file, 0.2 MB.

**TABLE S3**, XLSX file, 2.9 MB.

**TABLE S4**, XLSX file, 17.6 MB.

**TABLE S5**, XLSX file, 3.4 MB.

**TABLE S6**, XLSX file, 0.1 MB.
**TABLE S7**, XLSX file, 0.1 MB.
**TABLE S8**, XLSX file, 0.04 MB.
**TABLE S9**, XLSX file, 0.2 MB.

## ACKNOWLEDGMENTS

We thank Steven Verbruggen from the Lab of Bioinformatics and Computational Genomics (BioBix) at Ghent University (Ghent, Belgium) for his help with running REPARATION.

This work was supported by the European Research Council (ERC) under the European Union's Horizon 2020 research and innovation program (PROPHECY grant agreement no. 803972 to P.V.D.).

We declare that we have no conflicts of interest with the contents of this article.

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
