## [Reviewer comments · mSystems]

Lost and found: re-searching and re-scoring proteomics data aids genome annotation and improves proteome coverage

Patrick Willems, Igor Fijalkowski, and Petra Van Damme

Corresponding Author(s): Petra Van Damme, Ghent University

Review Timeline:

Submission Date:	August 26, 2020
Editorial Decision:	September 22, 2020
Revision Received:	October 8, 2020
Accepted:	October 9, 2020

Editor: Gilles van Wezel

Reviewer(s): Disclosure of reviewer identity is with reference to reviewer comments included in decision letter(s). The following individuals involved in review of your submission have agreed to reveal their identity: Leonard J. Foster (Reviewer #1); Chao Du (Reviewer #3)

Transaction Report:

DOI: <https://doi.org/10.1128/mSystems.00833-20>

Manuscript id: mSystems00144-20

Title: Re-searching and re-scoring proteomics data aids genome annotation and improves proteome coverage

To Prof. Gilles van Wezel, Editor of mSystems,

Dear Prof. van Wezel,

On behalf of my co-authors, I would like to thank you for sending the comments on our manuscript entitled 'Re-searching and re-scoring proteomics data aids genome annotation and improves proteome coverage'. We appreciate the opportunity to adequately address the criticisms raised by the two reviewers. Since the general concept of our manuscript was generally appreciated and one of the two reviewers even found our findings very exciting, we now made the concept of our revised manuscript clearer. Since clearly an awareness is needed for the value of proteogenomics-aided genome (re-)annotation, we believe that our manuscript makes a very significant contribution to the field and to be of general interest for the microbiology community viewing the importance of accurate bacterial genome annotation.

Further, the improved annotations obtained using our stringent workflow is amongst others clearly exemplified by the identification of 8 previously unannotated small ORF encoded polypeptides or SEPs, a category of proteins generally underrepresented in proteomic screens and which remain undetectable even in very recent proteomics efforts in pursue of identifying and characterizing (unannotated) small bacterial proteins using state-of-the-art MS instruments but standard workflows (<https://doi.org/10.1101/2020.05.26.116038>).

In the 'Response to reviewers' document, a point-by-point response to the specific concerns raised in the reviews and an indication of the respective changes made to the original manuscript is provided. Below, the editorial suggestions made were commented upon.

We hope that the revised manuscript can now be found acceptable for publication in mSystems and are looking forward to a favorable reply.

Sincerely yours,

Van Damme Petra

Editorial comments

I have received the reviews of your manuscript. While your paper addresses an interesting question, the reviewers stated several concerns about your study and did not recommend publication in mSystems. As you will see there are very mixed responses from the reviewers. I have carefully read the manuscript myself and I agree with reviewer #2 that a major rehaul is needed to make this paper suitable for mSystems. I do appreciate the interesting concept but the way it is presented it would be more suited for a dedicated proteomics journal. As an example, an important application is clearly the purpose of genome re-annotation and finding extended N-termini or frame shifts. However, for that, better comparison with existing methods is needed. As you know, at mSystems we are committed to making rapid final decisions. Because it appears that addressing the reviewers' concerns will require a significant amount of additional work that would delay the ultimate outcome, my decision at this time is to reject the manuscript.

If you feel that you can address the criticisms of the reviewers, you may submit a revised manuscript to mSystems as a new submission, which will be assigned a new manuscript number and receipt date. Please note the previous manuscript number and my name in the cover letter. Provide point-by-point responses to the issues raised by the reviewers in a file named "Response to Reviewers," not in your cover letter. Upload a compare copy of the manuscript (without figures) as a "Marked-Up Manuscript" file. In the response file, specify with page and line numbers where the revisions have been made in the marked-up manuscript.

While the first reviewer is clearly supportive of publication, the study was likely not recommended for publication based on the comments raised by Reviewer 2 while we felt that the reviewer missed out some important points already included in the manuscript. While we understand the editor's viewpoint that this paper – generally acknowledged of being (very) interesting - might be more suited for a dedicated proteomics journal, with the reformatting of the revised manuscript, we now made the manuscript clearer in its intent and highlight the clear benefits of our stringent workflow over existing routine methods by providing a comparison. We now demonstrated better the added value of re-scoring and re-searching in terms of proteogenomic ORF delineation, and the need for class-specific FDR scoring. Overall, we are convinced that the general concept of this important work is of high importance for scientists working in the field of molecular microbiology and for the annotation of bacterial genomes in general.

Response to Reviewers

Reviewer #1 (Comments for the Author):

This is a very exciting manuscript that describes a pipeline for very fine-grained annotation of bacterial genomes from proteomic data. I believe that the approach the authors have used here is a suitable one but I must admit that I got lost in the details more than a few times. To this end, I think that the manuscript could really be enhanced by an additional figure, and possibly some detailed step-by-step instructions. For the figure, I think that something summarizing the results of each step, how the steps are logically linked and how all the steps lead to the final results, would help. This is covered to some extent in Fig. 5 but the whole process is still really hard to follow. To this end, some step-by-step instructions would REALLY help enable others to apply these methods. E.g., take all the spectra and search against NR peptide database with MQ, then take the spectra that do not match and... etc.

We would like to thank the reviewer for the appraisal of our work. As suggested by the reviewer, we now refer in the text to an additional new figure posting by the third-party service Figshare; <https://doi.org/10.6084/m9.figshare.12847904> (and see figure below), that illustrates in more detail the iterative search process of our pipeline (MS-GF+ search, PSM scoring feature table, Percolator processing and spectrum cleaning), and this for an example scan. We refer to this figure when explaining the concept of re-searching and re-scoring on page 12 lines 244-246.

We believe to have summarized the results of each iterative search conveniently in Figure 2 and referred to the corresponding peptide/PSM identification numbers in Supplemental Table S2 (or all corresponding data in Supplemental Table S3). Note that other steps of the pipeline such as cleaned MS/MS spectra, Percolator feature files, etc. are large supplementary files that are largely unreadable/presentable to the reader. We do however now elaborate on the added value of the advanced proteomic pipeline steps (i.e. re-searching and re-scoring), showing that 8 out of 66 high-confident ORFs (<https://doi.org/10.6084/m9.figshare.12850148>, panel C) were exclusively identified due to these pipeline refinements (page 18, lines 372-379). We would also like to refer to our answer in response to a comment made by reviewer 2 for more details.

We further expanded the step-by-step experimental procedures for the re-searching and re-processing. However, we now moved this section to the Supplementary Methods to improve the readability and keep the focus of the work on how this advanced proteomic searching aids bacterial genome annotation and improves proteome coverage.

New Supplemental Figure (see <https://doi.org/10.6084/m9.figshare.12847904>). **Example of re-searching and re-scoring for scan 64,409 in sample OD 0.4 rep #1.** Proteomic data was searched against the constructed proteogenomic database (see Methods) using MS-GF+. MS-GF+ outputted Percolator PSM scoring features were merged with additional features (see Table S1 for full overview), including Pearson correlation to MS2PIP-predicted spectra as estimated by reScore (1). Percolator was used for re-scoring all PSMs in a class-specific manner separately for novel and annotated peptides. In this case, the annotated peptide 'RVVVGLLLGEVIR' was a significant PSM with Percolator peptide Q-value of $1.31e^{-15}$. As for all PSMs with a Q-value ≤ 0.01 , the original spectrum was cleaned from matching b- and y-ions to identify co-fragmenting peptides in additional search rounds, similar as described by Shteynberg *et al.* (2). In these iterative search rounds (3 in total), similar Percolator processing was used and led here to the identification of 'SAEALQWDLSFR' and 'AGLPAGVLNLVQGGR'.

Reviewer #2 (Comments for the Author):

The manuscript by Van Damme and team entitled, Lost and Found: Re-searching and re-scoring proteomics data aids the discovery of bacterial proteins and improve proteome coverage, discuss a proteomics-based pipeline to improve the annotations of bacterial genomes. While there are many important points and ideas in this manuscript, it's not well representative of the literature where proteomics has been used to reannotate bacterial species including Salmonella the primary model for the pipeline development (see Pubmed: 14730672, and 17690205).

Again, we would like to thank the reviewer for the appreciation of our work. We agree with the reviewer that several proteogenomics efforts have already been reported to aid in bacterial genome annotation and now referenced these studies in the manuscript. We indeed use *Salmonella* as primary model, given our lab's focus and availability of high-quality ribo-seq data as an independent quality metric for novel ORF translation (see below). However, we did demonstrate the general applicability and promise of (re-)analyzing proteomics data for the less characterized bacteria *Deinococcus radiodurans*. So rather than providing a dedicated *Salmonella* re-annotation effort, our pipeline rationale is to aid in novel annotation in conjunction with automated bacterial genome annotations.

We now referenced the pioneering paper from Jaffe *et al.* that introduces proteogenomics (page 4, line 82). While the second study of which note by the reviewer mainly focuses on PTMs, and thus not so much on gene annotation, this study again emphasizes the importance of complementing every genome sequencing project with a proteomics effort as to significantly improve both genome and proteome annotations for a reasonable cost, meaning that optimized proteogenomics workflows are indispensable.

This manuscript needs to be much clearer in its intent. Is it organized primarily around annotation improvements? Then a comparison or understanding of the challenges of annotation needs to be presented. If it is the recovery of chimeric peptides as the goal, that has been published previously by the corresponding author (including data that appears in a supplemental figure -E. coli- that has no corresponding methods) and the differences are unclear. There is a serious need for tables for organization and comparisons of numbers. If several values change with different steps those should be tabulated by different steps. The manuscript and ideas are often disrupted by information better left elsewhere. Can comparisons of respective refinements by the pipeline be shown? How many new N-termini, validation of small proteins, etc. That might show future researchers what steps have the most promise for improvement or compare their methodologies.

Foremost, to make the manuscript clearer in its intent, we now changed the title from 'Re-searching and re-scoring proteomics data aids the discovery of bacterial proteins and improves proteome coverage' to 'Re-searching and re-scoring proteomics data aids genome annotation and improves proteome coverage'. Hence, our main purpose is to demonstrate that by advanced (re-)analysis of public/newly generated data, a higher proteome coverage and sensitive detection of unannotated ORFs can be achieved. This intent is now clearly stated in the manuscript. As elaborated upon in a recent review of our lab (3), we now specified some of the challenges associated with annotation (page 4 lines 72-74) and referenced our recent work which highlights annotation challenges.

As indicated in the response to Reviewer 1, we moved some 'disruptive' technical aspects of the proteomic re-scoring and re-searching to the Supplemental Methods to improve readability and focus of the work. Further, while recovery of chimeric peptides was clearly not the overarching goal, the

implementation of this iterative search strategy combined with Percolator post-processing was shown to increase the number of confident peptide identifications. In addition, such co-fragmenting peptides can indicate low abundant peptide species that could be generally missed in routine proteomics analysis, enhancing the chance of identifying translation products of unannotated ORFs. As indicated in the original manuscript, a similar iterative search of a 'cleaned' spectrum for MS/MS spectra that potentially contain co-fragmented peptides can also be performed in MaxQuant (4), but was not published by us previously.

Further we would like to point out to the reviewer that no experimental *E. coli* data is included in the manuscript or is shown in (supplementary) figures. Part of the *E. coli* prfB protein sequence is simply shown for the purpose of orthologues comparison (correct gene annotation in *E. coli*) and demonstrating a spurious ORF annotation in case of *Salmonella enterica*.

As pointed out by the reviewer, we now more clearly analyzed the effect of the respective refinements to the identification of novel ORFs (similar comment as Reviewer #1 and referring to Figure 5C). As was mentioned in the manuscript before, the auxiliary and combined feature sets deliver additional novel peptide identifications (Figure 5A). We now determined the beneficial effect of these feature sets on the 66 high-confident ORFs (Figure 5C) (marked changes page 18, lines 372-379). This shows that 7 out of 66 novel high-confident ORFs (10.7%) were identified solely due to using the auxiliary and/or feature sets. Note that for an additional 13 high-confident ORFs, the feature sets were able to identify additional peptides next to a peptide identified by default MS-GF+Percolator processing. Next to using extended PSM scoring features, also one high-confident ORF, the N-terminal extension of RuvB, was solely identified due to iterative searching. These 8 ORFs now identified due to re-scoring or re-searching were highlighted in bold in a new version of Figure 5C. We believe this convincingly demonstrates the added value of the refinements made by the pipeline, and was shortly summarized in the results section accordingly.

If improved genome annotation by this complex multi-step analysis is the goal then a comparison to simpler methodologies (6 frame translation and filtering based on standard criteria) is necessary and literature published examples. The improved annotations are likely nominal over a straight 6-frame translation and even worse if the next best PSM of excellent score is used. The work would be better served either dropping the new bacterium or using another Salmonella dataset. The new bacterium adds nothing to the clarity of this manuscript.

Currently, we validated the performance of our pipeline by comparing identified annotated peptides with those identified by a routine MaxQuant search of which the results are provided in Figure 2C. As elaborated upon in the manuscript, a greater proteome coverage can be attained using our iterative search strategy.

However, we certainly agree with the reviewer that we should include an additional comparison for the six frame translation searches with standard criteria. To this end, we compared our refinements (i.e. re-scoring and re-searching) to the default MS-GF+Percolator already described in the paper. Importantly, we ran Percolator without class-specific FDR scoring ('combined FDR' scoring for annotated and novel peptides), as this is regrettably still often performed in recent proteogenomic searches (5, 6), against posed proteogenomic criteria. As a benchmark, we plotted the available *Salmonella* ribo-seq data per peptide. More precisely, we calculated the reads per kilobase per million (RPKM) for the genomic region corresponding to the peptide. We then plotted the relative proportions of identified peptides with a RPKM > 10, < 10 or 0 in a novel panel of Figure 4 (Figure 4D). As now described in the results section on page 13-14 (lines 305-317), the proportion of peptides with high ribo-seq translation evidence (RPKM > 10) varies

from 30 to 40% if using a combined FDR, while above 80% using a class-specific FDR 5% threshold for novel peptides. While we cannot claim peptides with a RPKM of 0 to be false positives, it clearly shows that class-specific FDR scoring favors the identification of unannotated translated peptides – which are intuitively desired. The class-specific scoring however identifies less peptides, but here the combined feature set re-scoring can identify for instance 119 novel peptides with a RPKM > 10 in the first search compared to 100 novel peptides by default MS-GF+ Percolator processing (Figure 4D). Whereas this benchmark was possible due to high quality ribo-seq data available for *Salmonella*, this is for instance not the case for *Deinococcus*. Without such orthogonal evidence, combined FDR scoring strategies thus pose a certain danger for proteogenomic-based annotation. We emphasize this in the discussion as a must if proteogenomic pipelines would be used in conjunction with automated *in silico* annotation. We previously illustrated the promise of such approach by using *Deinococcus* in the last paragraph with publicly available MS data – which is in our opinion clearly of relevance. Note that we selected *Deinococcus* initially due to the relatively poor MS sampling of its clade, as described on lines 450 to 455 on page 21.

Figure 4D. Ribo-seq coverage for annotated and novel peptides identified in the first and second search (left and right, respectively) using different feature sets for combined FDR or class-specific FDR estimation (top and bottom, respectively). Ribo-seq Reads Per Kilobase of transcript per Million reads mapped (RPKM) were calculated for genomic regions encoding the respective peptide, distinguishing highly translated (RPKM > 10), lowly translated (RPKM < 10) and peptide genomic regions without ribosome footprints (RPKM = 0).

Further, the improved annotations obtained using our stringent workflow is amongst others clearly exemplified by the identification of 8 previously unannotated small ORF encoded polypeptides or SEPs (Fig. 5C and Fig. R1), a category of proteins generally underrepresented in proteomic screens and which remain undetectable even in very recent proteomics efforts in pursue of identifying and characterizing (unannotated) small bacterial proteins using state-of-the-art MS instruments but standard workflows (<https://doi.org/10.1101/2020.05.26.116038>).

A

SEP	Genome position	Length	Peptides
SEP1	pCol1B9:42,148-42,396 ²	83 AA	LELIGFR
SEP2	pCol1B9:11,613-11,870 ²	86 AA	VFSLSYEQLTR
SEP3	Chr:2,893,731-2,893,988 ^{1,2}	86 AA	DLNSQIINITTNR
SEP4	Chr:2,969,532-2,969,708 ²	59 AA	MFTPGDIVQPR
SEP5	Chr:2,071,925-2,072,146 ²	74 AA	SHGYTLQHVAK
SEP6	pSLT:6,736-6,963 ²	76 AA	AVALPENVK RVEVIAVGR
SEP7	Chr:901,221-901,347 ^{1,2}	42 AA	VFSHSPFK FEINPVNNR
SEP8	Chr:3,222,490-3,222,564	25 AA	MFEINPVNNR IQDLTER

B**C****D****E****F**
Figure R1 | Newly identified small ORF-encoded polypeptides (SEPs, < 100 amino acids) with proteomics evidence, ranked from SEP1 to SEP8 (**A**). (**B-F**) SEP1 to SEP5 were further illustrated by a genome browser track displaying strand-specific ribo-seq coverage (left panels) and corresponding annotated MS/MS spectra with b, y and precursor ions (blue, red and purple, respectively).

Another comment:

1) The Salmonella proteomics data production should be in the made portion of the article. The reader should be able to refer to it without looking at supplementary material.

As suggested by the reviewer, we now included the methods parts on 'Bacterial cultivation' and 'Proteomic sample preparation', originally in supplemental material, in the 'material and methods' section of the main manuscript. Further, a brief description of the proteomics samples analyzed was provided in the text.

REFERENCES

1. AS CS, Bouwmeester R, Martens L, Degroev S. 2019. Accurate peptide fragmentation predictions allow data driven approaches to replace and improve upon proteomics search engine scoring functions. *Bioinformatics* 35:5243-5248.
2. Shteynberg D, Mendoza L, Hoopmann MR, Sun Z, Schmidt F, Deutsch EW, Moritz RL. 2015. reSpect: software for identification of high and low abundance ion species in chimeric tandem mass spectra. *J Am Soc Mass Spectrom* 26:1837-47.
3. Fijalkowska D, Fijalkowski I, Willems P, Van Damme P. 2020. Bacterial riboproteogenomics: the era of N-terminal proteoform existence revealed. *FEMS Microbiol Rev* doi:10.1093/femsre/fuaa013.
4. Cox J, Neuhauser N, Michalski A, Scheltema RA, Olsen JV, Mann M. 2011. Andromeda: a peptide search engine integrated into the MaxQuant environment. *J Proteome Res* 10:1794-805.
5. Miravet-Verde S, Ferrar T, Espadas-Garcia G, Mazzolini R, Gharrab A, Sabido E, Serrano L, Lluch-Senar M. 2019. Unraveling the hidden universe of small proteins in bacterial genomes. *Mol Syst Biol* 15:e8290.
6. Herbst FA, Goncalves SCL, Behr T, McIlroy SJ, Nielsen PH. 2019. Proteogenomic Refinement of the *Neomegalonema perideroedes*(T) Genome Annotation. *Proteomics* 19:e1800330.

September 22, 2020

Prof. Petra Van Damme
Ghent University
Department of Biochemistry and Microbiology
K. L. Ledeganckstraat 35
Ghent 9000
Belgium

Re: mSystems00833-20 (Lost and found: re-searching and re-scoring proteomics data aids genome annotation and improves proteome coverage)

Dear Prof. Petra Van Damme:

Thanks for the resubmission of your paper. As you will see, one previous reviewer now recommends acceptance while a second (new) reviewer is positive and has a few comments for your consideration. Before I can accept the paper, I would like to ask you to address these comments appropriately.

Below you will find the comments of the reviewers.

To submit your modified manuscript, log onto the eJP submission site at <https://msystems.msubmit.net/cgi-bin/main.plex>. If you cannot remember your password, click the "Can't remember your password?" link and follow the instructions on the screen. Go to Author Tasks and click the appropriate manuscript title to begin the resubmission process. The information that you entered when you first submitted the paper will be displayed. Please update the information as necessary. Provide (1) point-by-point responses to the issues raised by the reviewers as file type "Response to Reviewers," not in your cover letter, and (2) a PDF file that indicates the changes from the original submission (by highlighting or underlining the changes) as file type "Marked Up Manuscript - For Review Only."

Due to the SARS-CoV-2 pandemic, our typical 60 day deadline for revisions will not be applied. I hope that you will be able to submit a revised manuscript soon, but want to reassure you that the journal will be flexible in terms of timing, particularly if experimental revisions are needed. When you are ready to resubmit, please know that our staff and Editors are working remotely and handling submissions without delay. If you do not wish to modify the manuscript and prefer to submit it to another journal, please notify me of your decision immediately so that the manuscript may be formally withdrawn from consideration by mSystems.

Sincerely,

Gilles van Wezel

Editor, mSystems

Journals Department
Reviewer comments:

Reviewer #1 (Comments for the Author):

The authors have adequately addressed my prior concerns.

Reviewer #2 (Comments for the Author):

The research article "Lost and found: re-searching and re-scoring proteomics data aids genome annotation and improves proteome coverage" by Patrick Willems, Igor Fijalkowski, and Petra Van Damme presents a bioinformatics pipeline that is likely to improve proteomics data analysis and also offer advantages in discovering new ORFs in the genomes. The novelty of this research lies in combining bioinformatics tools for the analysis of LC-MS/MS data. These tools evaluate peptide identifications (peptide-spectrum matches, PSMs) in multiple dimensions that utilize information from LC-MS/MS runs. This including data sets like retention times, which were previously often disregarded. Secondly, the integration of genomics into the pipeline shows how genome annotation can be improved. Finally, the pipeline can be applied in already published datasets, further underlining the usefulness of the methods.

I also have several comments for the authors' consideration.

[1] Improvements of the genome annotation and proteome coverage are expected to be scarce and random if they are just based on a single proteomics experiment. After all, the coverage will be relatively low, say between 10-30% of the proteome / genome. This has repercussions for the accuracy of the improved annotation. Thus, the pipeline needs to be widely applied to many proteogenome experiments to expand its value.

[2] the authors show in their article that this pipeline involves manual screening and evaluation. While I would be happy to use it myself, automated tools should be developed to increase their applicability, as now scientists cannot rely fully on this pipeline to do "targeted" research. A comment should be made to this effect, with suggestion as to what would be aimed for.

[3] The field is moving towards combining 'omics technologies, such as combination of quantitative proteomics with high-resolution LC-MS/MS experiments. This will empower systems biology as a

prevailing tool in biology research. It would be good to expand a bit on the position of this work within the broader 'omics field in the Discussion Section.

Response to Reviewers

Reviewer #2 (Comments for the Author):

The research article "Lost and found: re-searching and re-scoring proteomics data aids genome annotation and improves proteome coverage" by Patrick Willems, Igor Fijalkowski, and Petra Van Damme presents a bioinformatics pipeline that is likely to improve proteomics data analysis and also offer advantages in discovering new ORFs in the genomes. The novelty of this research lies in combining bioinformatics tools for the analysis of LC-MS/MS data. These tools evaluate peptide identifications (peptide-spectrum matches, PSMs) in multiple dimensions that utilize information from LC-MS/MS runs. This including data sets like retention times, which were previously often disregarded. Secondly, the integration of genomics into the pipeline shows how genome annotation can be improved. Finally, the pipeline can be applied in already published datasets, further underlining the usefulness of the methods.

We would like to thank the reviewer for the appreciation of our work.

I also have several comments for the authors' consideration. Improvements of the genome annotation and proteome coverage are expected to be scarce and random if they are just based on a single proteomics experiment. After all, the coverage will be relatively low, say between 10-30% of the proteome / genome. This has repercussions for the accuracy of the improved annotation. Thus, the pipeline needs to be widely applied to many proteogenome experiments to expand its value.

First we would like to emphasize that the original Salmonella proteomics data included already covered 3 different (closely related) growth conditions (i.e. 3 exponential growth phases; OD₆₀₀ 0.2, OD₆₀₀ 0.4, OD₆₀₀ 0.6) which were all sampled in biological triplicates (i.e., 9 total proteome shotgun samples). By itself, this high-depth coverage dataset resulted in the identification of 3202 out of the 4670 annotated S. Typhimurium SL1344 proteins (69% coverage), thereby reproducing the translated proteome as deduced from ribosome profiling (1) and making it one of the most highly covered S. Typhimurium datasets reported to date. The 69% proteome coverage is thus a much higher degree of coverage as the 10-30% noted by the reviewer, overall making these very suitable datasets for proteogenomics.

Nonetheless, we followed the suggestion of the reviewer to apply the pipeline to additional proteomics data aiming at further improving genome annotation and proteome coverage. For this, a complex proteome of Salmonella grown in vitro under 10 different (infection relevant) growth conditions as reported in (2) and selected based on the complementarity in RNA expression data (3) - and thus likely to provide a more comprehensive proteome coverage - was analyzed using our proteogenomics pipeline in an automated manner.

More specifically, we performed an offline RP-HPLC pre-fractionation of a digest of a complex proteome mixture obtained from mixing equal proteome amounts of 10 different (infection relevant) conditions. For this, Salmonella were grown to early exponential growth phase (EEP; OD₆₀₀ 0.1), mid exponential growth phase (MEP; OD₆₀₀ 0.3), late exponential growth phase (LEP; OD₆₀₀ 1.0), early stationary phase (ESP; OD₆₀₀ 2.0) and late stationary phase (LSP; OD₆₀₀ 2.0 + 6 h of extra growth). Besides, environmental shocks in Luria Bertani medium were performed on MEP-grown bacteria by the addition of NaCl to a final concentration of 0.3 M and growth was allowed to continue for 10 min or, in case of anaerobic shock, for an additional 30 min without agitation in a filled and tightly screwed 50 mL Falcon tube. For growth in variants of PCN

minimal medium (4), overnight grown LB cultures were washed twice in PCN medium before resuspension at $O.D_{600}$ 0.02. Cells were grown in SPI2-inducing PCN or low magnesium SPI2-inducing PCN. The nitric oxide shock conducted in PCN (InSPI2) was performed at OD_{600} 0.3 by addition of the nitric oxide donor spermine NONOate to a final concentration of 250 μ M for 20 min (nitric oxide shock (InSPI2)) (5).

Besides the identification of 3120 annotated protein IDs originating from 25,361 unique peptides (numbers in line with those obtained when searching the 9 total shotgun proteome samples (i.e., the 3 exponential growth phases sampled in biological triplicates), this analysis resulted in the identification of 40 novel proteogenomic peptides overlapping with our previous analysis. Besides, 9 non-redundant proteogenomic peptides identified in this analysis did not overlap with any of the 193 previously reported proteogenomic peptides (i.e., proteogenomic peptides identified in the 9 total proteome samples). More specifically, 6 of these novel identifications, confirmed the translation products of newly identified ORFs previously classified as being high confident in the manuscript, providing an even higher confidence for these proteogenomic identifications. The other 3 novel identifications matched two N-terminal protein extensions (one of which represents a 2 amino acids extension), next to an N-terminally truncated proteoform (see new Supplemental Table 7C). Upon inspection of our previously reported Ribo-seq based Salmonella annotations, expression of the two N-terminally extended proteoforms was additionally confirmed (data not shown) (1, 6). The relative lower number of novel proteogenomic peptides identified in these analysis (i.e., 49 versus 193) might be due to the increased complexity of the sample upon mixing of the different proteomes and therefore lower sensitivity to identify new proteogenomic peptides of relative lower abundance. Nonetheless, we clearly demonstrate that inclusion of diverse proteogenomic datasets improves bacterial genome annotation and proteome coverage and now acknowledge this potential and include a description of this data in the manuscript in support of this statement.

Overall, this data confirms the validity of our proteogenomic findings and clearly demonstrates the wide applicability and value of our proteogenomic workflow for bacterial genome annotation.

[2] The authors show in their article that this pipeline involves manual screening and evaluation. While I would be happy to use it myself, automated tools should be developed to increase their applicability, as now scientists cannot rely fully on this pipeline to do "targeted" research. A comment should be made to this effect, with suggestion as to what would be aimed for.

The analysis performed on the complex proteome samples when mixing the proteomes of Salmonella grown in vitro under 10 different growth conditions was performed when using our workflow in an automated fashion. The results were found in agreements with our previous proteogenomic analysis performed (see also our answer in response of comment 1). Besides the novel hits identified, an increased proteome coverage for some of the previously identified proteogenomic identifications could be found, again confirming previously identified proteogenomic hits. Viewing the high-confidence of the novel proteogenomic peptide identifications (e.g. by the use of strict class-specific FDR scoring), it is important to note that our workflow can be applied irrespective of manual inspection. Manual inspection was originally done to support our findings. More specifically, as shown in Figure 4D, identified novel peptides show a high ribo-seq signal – a solid independent indicator of protein-coding potential. Also for the novel proteogenomic analysis included, expression of the two N-terminally extended proteoforms could be confirmed based on available ribo-seq data (1, 6). We now more clearly highlight the value of using our optimized proteogenomic workflow.

[3] The field is moving towards combining 'omics technologies, such as combination of quantitative proteomics with high-resolution LC-MS/MS experiments. This will empower systems biology as a prevailing tool in biology research. It would be good to expand a bit on the position of this work within the broader 'omics field in the Discussion Section.

As suggested by the reviewer, we now also expanded on the position of our work in a broader context in the discussion section.

References

1. E. Ndah *et al.*, REPARATION: ribosome profiling assisted (re-)annotation of bacterial genomes. *Nucleic Acids Res* **45**, e168 (2017).
2. C. Kroger *et al.*, An infection-relevant transcriptomic compendium for Salmonella enterica Serovar Typhimurium. *Cell Host Microbe* **14**, 683-695 (2013).
3. S. Srikumar *et al.*, RNA-seq Brings New Insights to the Intra-Macrophage Transcriptome of Salmonella Typhimurium. *PLoS pathogens* **11**, e1005262 (2015).
4. S. Lober, D. Jackel, N. Kaiser, M. Hensel, Regulation of Salmonella pathogenicity island 2 genes by independent environmental signals. *International journal of medical microbiology : IJMM* **296**, 435-447 (2006).
5. T. J. Bourret *et al.*, Nitric oxide antagonizes the acid tolerance response that protects Salmonella against innate gastric defenses. *PLoS One* **3**, e1833 (2008).
6. A. Giess *et al.*, Ribosome signatures aid bacterial translation initiation site identification. *BMC Biol* **15**, 76 (2017).

October 9, 2020

Prof. Petra Van Damme
Ghent University
Department of Biochemistry and Microbiology
K. L. Ledeganckstraat 35
Ghent 9000
Belgium

Re: mSystems00833-20R1 (Lost and found: re-searching and re-scoring proteomics data aids genome annotation and improves proteome coverage)

Dear Prof. Petra Van Damme:

Your manuscript has been accepted, and I am forwarding it to the ASM Journals Department for publication. For your reference, ASM Journals' address is given below. Before it can be scheduled for publication, your manuscript will be checked by the mSystems senior production editor, Ellie Ghatineh, to make sure that all elements meet the technical requirements for publication. She will contact you if anything needs to be revised before copyediting and production can begin. Otherwise, you will be notified when your proofs are ready to be viewed.

Sincerely,

Gilles van Wezel
Editor, mSystems

Journals Department
Table S2: Accept

Table S7A-C: Accept

Table S6: Accept

Table S1: Accept

Table S9A-B: Accept

Table S3: Accept

Supplemental Methods: Accept

Table S5: Accept

Table S4A-B: Accept

Table S8: Accept